# PARAMETRIC AUGMENTATION FOR TIME SERIES CONTRASTIVE LEARNING

**Xu Zheng[1], Tianchun Wang[2], Wei Cheng[3], Aitian Ma[1], Haifeng Chen[3], Mo Sha[1], Dongsheng Luo[1]✉**

[1]School of Computing and Information Sciences, Florida International University, US
[2]College Information Sciences and Technology, The Pennsylvania State University, US
[3]NEC Laboratories America, US

```
{xzhen019,aima,msha,dluo}@fiu.edu
tkw5356@psu.edu,
{weicheng,haifeng}@nec-labs.com
```

## ABSTRACT

Modern techniques like contrastive learning have been effectively used in many areas, including computer vision, natural language processing, and graph-structured data. Creating positive examples that assist the model in learning robust and discriminative representations is a crucial stage in contrastive learning approaches. Usually, preset human intuition directs the selection of relevant data augmentations. Due to patterns that are easily recognized by humans, this rule of thumb works well in the vision and language domains. However, it is impractical to visually inspect the temporal structures in time series. The diversity of time series augmentations at both the dataset and instance levels makes it difficult to choose meaningful augmentations on the fly. In this study, we address this gap by analyzing time series data augmentation using information theory and summarizing the most commonly adopted augmentations in a unified format. We then propose a contrastive learning framework with parametric augmentation, AutoTCL, which can be adaptively employed to support time series representation learning. The proposed approach is encoder-agnostic, allowing it to be seamlessly integrated with different backbone encoders. Experiments on univariate forecasting tasks demonstrate the highly competitive results of our method, with an average 6.5% reduction in MSE and 4.7% in MAE over the leading baselines. In classification tasks, AutoTCL achieves a 1.2% increase in average accuracy. The source code is available at https://github.com/AslanDing/AutoTCL.

## 1 INTRODUCTION

Time series data is complex and high-dimensional, making it more difficult to gather the label than images or languages. This property hinders the deployment of powerful deep learning methods, which typically require a large amount of labeled data for training(Eldele et al., 2021). Self-supervised learning is a promising solution due to its capacity to learn from unlabelled data. Self-supervised learning methods learn a fixed-dimension embedding of the time series data that preserves its inherent features with better transferability and generalization capacity. A representative framework, contrastive learning, has been successful in representation learning for various types of data including vision, language, and graphs(Chen et al., 2020; Xie et al., 2020; You et al., 2020). These methods train an encoder to map instances to an embedding space where similar instances (positive pairs) are easily distinguished from dissimilar ones (negative pairs). As a result, model predictions are unaffected by minor noise introduced into the inputs or hidden states. As a key component, data augmentation such as jittering, scaling, permutation, and subsequence extraction (Fan et al., 2020; Wen et al., 2021), is usually adopted to produce positive pairs.

Recently, some efforts have been made to develop contrastive learning methods for time series data (Eldele et al., 2021; Franceschi et al., 2019; Fan et al., 2020; Tonekaboni et al., 2021). However,

---

✉Corresponding author.

due to the diversity and variability of real-world time series data, it is challenging to apply a general augmentation technique to all datasets. As a result, current approaches to contrastive learning for time series data frequently need particular data augmentation techniques that are guided by domain knowledge and necessitate trial and error to identify the most appropriate augmentations. Attempts have recently been made to study the theory behind adaptive augmentation selection for contrastive learning (Tian et al., 2020; Suresh et al., 2021; Xu et al., 2021). Good augmentations, according to InfoMin (Tian et al., 2020), produce label-preserving views with less shared information. They discover the best perspectives through adversarial training in unsupervised or self-supervised environments by adding an invertible flow-based generative model. The InfoMin principle performs well in the vision area and has been successfully applied to graph-structured data (Xu et al., 2021; Suresh et al., 2021; Yin et al., 2022; You et al., 2022). However, in a self-supervised environment, most existing studies soften the label-preserving property and place a more significant emphasis on enhancing diversity by reducing the exchange of information between different views. They frequently use stronger transformers as augmentations and undermine the semantics, which is inapplicable to time series data.

To accommodate various augmentation tactics for time series contrastive learning, we investigate the data augmentation for time series from the information theory perspective and provide a theoretically sound definition of good augmentations based on input factorization. We further present a contrastive learning framework with parametric augmentation, AutoTCL, to adaptively augment data for contrastive time series learning based on the proposed factorization technique, which can prevent ad-hoc decisions or laborious trial-and-error tuning. Specifically, we utilize a parametric neural network to learn to factorize an instance into two parts: the informative part and the task-irrelevant part. The informative component is then applied to a lossless transform function which keeps the instance's semantics. The adaptive transformation produces a prior mask for the input instance to generate workable positive samples. We demonstrate how the most effective time series data augmentation methods can be viewed as specialized forms of the suggested mask-based transformation. By including another random variable with adequate variance, the diversity of the augmented view is further increased. In order to learn representations through contrast, augmented pairs are then fed into a time series encoder along with randomly chosen negative pairings. Parameters in the factorization and transform functions are optimized in tandem with contrastive learning loss. Our main contributions are summarized as follows.

- We introduce a novel factorization-based framework to guide data augmentations for contrastive self-supervised learning without prefabricated knowledge.
- To automatically learn workable augmentations for time series data, we provide a straightforward yet effective instantiation that can handle a variety of frequently applied augmentations.
- With comprehensive experimental studies, we empirically verify the advantage of the proposed method on benchmark time series forecasting datasets. We achieve highly competitive performances with a 6.5% reduction in MSE, 4.7% in MAE on univariate forecasting, a 2.9% reduction in MSE, and 1.2% in MAE on multivariate forecasting. In classification tasks, our method achieves a 1.2% increase in average accuracy.

## 2 RELATED WORK

**Contrastive learning for time series.** Contrastive learning has been widely used in representation learning, achieving superior results across various domains (Chen et al., 2020; Xie et al., 2020; You et al., 2020). Recently, there have been efforts to apply contrastive learning to the time series domain (Khaertdinov et al., 2021; Oord et al., 2018; Franceschi et al., 2019; Fan et al., 2020; Eldele et al., 2021; Tonekaboni et al., 2021; Yue et al., 2022; Yang & Hong, 2022). In (Franceschi et al., 2019), Franceschi et al. utilize subsequences to generate positive and negative pairs. TNC uses a debiased contrastive objective to make sure that in the representation space, signals from the local neighborhood are distinct from those that are not neighbors (Tonekaboni et al., 2021). TS2Vec uses hierarchical contrastive learning to acquire a representation for each time stamp (Yue et al., 2022). TF-C utilizes the distance between time and frequency components as the self-supervised signal for representation learning. Each component is independently optimized by contrastive estimation (Zhang et al., 2022). In (Nonnenmacher et al., 2022), the authors introduce an approach that incorporates expert knowledge into time-series representation learning using expert features, surpassing existing

methods in unsupervised and semi-supervised learning on real-world datasets. CLUDA (Ozyurt et al., 2023), a novel framework for unsupervised domain adaptation of time series data, utilizes contrastive learning to learn contextual representations that preserve label information, achieving state-of-the-art performance in time series unsupervised domain adaptation.

**Adaptive data augmentation.** Data augmentation is a crucial aspect of contrastive learning. Previous studies have shown that the choice of optimal augmentation methods depends on the specific task and dataset being used (Chen et al., 2020; Fan et al., 2020). Some studies have explored the adaptive selection of augmentation methods in the visual domain Tamkin et al. (2021); Cubuk et al. (2019); Hataya et al. (2020); Li et al. (2020); Tian et al. (2020); Rommel et al. (2022); Aboussalah et al. (2023); Ho et al. (2019). For example, AutoAugment (Cubuk et al., 2019) uses a reinforcement learning method to search for the best combination of policies. Later, CADDA investigates a gradient-based class-wise method to support larger search spaces for EGG signals (Rommel et al., 2022). DACL adopts a domain-agnostic approach that does not rely on domain-specific data augmentation techniques (Verma et al., 2021). MetAug (Li et al., 2022) and Hallucinator (Wu et al., 2023) aim to generate augmentations in the latent space. In the contrastive learning frameworks, the InfoMin theory is applied to guide the selection of good views for contrastive learning in the vision domain (Tian et al., 2020), it further proposes a flow-based generative model to transfer images from natural color spaces into novel color spaces for data augmentation.

However, given the complexity of time series data, directly applying the InfoMin framework may not be suitable. Different from previous works, our focus is on the time series domain and we propose an end-to-end differentiable method to automatically learn the optimal augmentations for each time series instance.

**Time series forecasting.** Forecasting is an essential component of time series analysis, and various deep learning architectures have been employed in the literature, such as MLP (Ekambaram et al., 2023), Recurrent Neural Networks (RNNs) (Salinas et al., 2020; Oreshkin et al., 2020), Convolutional Neural Networks (CNNs) (Bai et al., 2018; Zhang et al., 2023b), Transformers (Li et al., 2019; Zhou et al., 2021; Lin et al., 2023; Yu et al., 2023; Nie et al., 2023; Zhang et al., 2023a), and Graph Neural Networks (GNNs) for this task (Cao et al., 2020). In contrast to these works, the aim of our research is to learn general representations for time series data that can be applied not only to forecasting but also to other tasks, such as classification. Additionally, the proposed framework is designed to be compatible with multiple types of architectures as encoders.

## 3 METHODOLOGY

In this section, we first describe the notations used in this paper. Then, we try to answer the following research questions. (1) What are the good views for contrastive learning in the self-supervised setting? (2) How to obtain good views for each time series instance for contrastive learning?

### 3.1 NOTATIONS

We use a $T \times F$ matrix to represent a time series instance $x$, where $T$ is the length of its sequence and $F$ is the dimension of features. With $F > 1$, $x$ is a multivariate time series instance. Otherwise, with $F = 1$, $x$ is a single variate instance. Self-supervised contrastive learning aims to learn an encoder $f$ that maps $x$ from $\mathbb{R}^{T \times F}$ to a vector space $\mathbb{R}^D$, where $D$ is the dimension of embedding vectors. In the paper, to distinguish random variables and instances, we use the Sans-serif style lowercase letters, such as x, to represent random time series variables, and italic lowercase letters, such as $x$, for real instances. Important notations are summarized in Appendix.

### 3.2 WHAT MAKES GOOD VIEWS FOR CONTRASTIVE SELF-SUPERVISED LEARNING?

In the literature, a well-accepted intuitive principle for view generation is that good views preserve the semantics and provide sufficient variances (Tian et al., 2020; Yin et al., 2022; Suresh et al., 2021). In the supervised setting, where training labels are available, the semantics of an instance is usually approximated with the label. On the other hand, semantics-preserving is much less explored in the more popular self-supervised learning. Moreover, while the semantics of images and natural

language sentences can be manually verified, the underlying semantics of time series data are not easily recognizable to humans. This makes it challenging, if not impossible, to apply strong yet faithful data augmentations to such data. To avoid the degenerate solutions caused by dismissing the semantics-preserving, InfoMin utilizes an invertible flow-based function, denoted by $g$, to generate a view $v$ for an input $x$ (Tian et al., 2020). Such that $x$ can be restored by $x = g^{-1}(v)$. However, from the information theory perspective, invertible functions fail to include extra variance to the original variable. Formally, we have the following property.

**Property 1.** *If view* $\mathsf{v}$ *is generated from* $\mathsf{x}$ *with an invertible function* $\mathsf{v} = g(\mathsf{x})$. *Then* $H(\mathsf{v}) = H(\mathsf{x}) = MI(\mathsf{x}; \mathsf{v})$, *where* $H(\mathsf{x}), H(\mathsf{v})$ *are entropy of variables* $\mathsf{x}$ *and* $\mathsf{v}$, *respectively;* $MI(\mathsf{v}; \mathsf{x})$ *is mutual information between* $\mathsf{v}$ *and* $\mathsf{x}$.

The detailed proof can be found in the Appendix. This property shows that the entropy of the augmented view, $H(\mathsf{v})$, is no larger than that of original data, $H(\mathsf{x})$, indicating that the existing data augmentation methods don't bring new information for input instances, which limits their expressive power for time series contrastive learning. To address the challenge and facilitate powerful self-supervised learning in the time series domain, we propose a novel factorized augmentation technique. Specifically, given an instance $x$, we assume that $x$ can be factorized into two parts, informative $x^*$ and task-irreverent part $\Delta x$. Formally,

$$x = x^* + \Delta x. \tag{1}$$

As the informative part, $x^*$ encodes the semantics of the original $x$. Motivated by the intuitive principle, we formally define good views for contrastive learning as follows.

**Definition 1** (Good View). *Given a random variable* $\mathsf{x}$ *with its semantics* $\mathsf{x}^*$, *a good view* $\mathsf{v}$ *for contrastive learning can be achieved by* $\mathsf{v} = \eta(g(\mathsf{x}^*), \Delta\mathsf{v})$, *where* $g$ *is an inverible function,* $\Delta\mathsf{v}$ *is a task-irrelevant noise, satisfying* $H(\Delta\mathsf{v}) \geq H(\Delta\mathsf{x})$, *and* $\eta$ *is an augmentation function that satisfies that* $g(\mathsf{x}^*) \to \mathsf{v}$ *is a one-to-many mapping.*

Intuitively, a good view, based on our definition, maintains the useful information in the original variable and at the same time, includes a larger variance to boost the robustness of encoder training. We theoretically show that the defined good view has the following properties.

**Property 2** (Task Agnostic Label Preserving). *If a variable* $\mathsf{v}$ *is a good view of* $\mathsf{x}$, *and the downstream task label* $\mathsf{y}$ *(although not visible to training) is independent to noise in* $\mathsf{x}$, *the mutual information between* $\mathsf{v}$ *and* $\mathsf{y}$ *is equivalent to that between raw input* $\mathsf{x}$ *and* $\mathsf{y}$, *i.e.,* $MI(\mathsf{v}; \mathsf{y}) = MI(\mathsf{x}; \mathsf{y})$.

**Property 3** (Containing More Information). *A good view* $\mathsf{v}$ *contains more information comparing to the raw input* $\mathsf{x}$, *i.e.,* $H(\mathsf{v}) \geq H(\mathsf{x})$.

Detailed proofs are given in the appendix. These properties show that in the self-supervised setting, adopting a good view for contrastive learning theoretically guarantees that we will not decrease the fidelity, regardless of the downstream tasks. Simultaneously, the good view is flexible to the choice of $\Delta\mathsf{v}$, meaning that strong augmentations may be utilized to incorporate enough diversity for training.

## 3.3 HOW TO ACHIEVE GOOD VIEWS?

The theoretical analysis suggests a factorized augmentation to preserve task-agnostic labels and improve the diversity of views. In this part, we introduce a practical instantiation to obtain good views based on parametric augmentations as demonstrated in Fig. 1. First, a factorization function $h : \mathbb{R}^{T \times F} \to \{0, 1\}^{T \times 1}$ is introduced to discover where are the informative parts in input. Formally[1],

$$\boldsymbol{h} = h(x), \qquad x^* = \boldsymbol{h} \odot x, \qquad \Delta x = x - x^*, \tag{2}$$

where $\boldsymbol{h}$ is the factorization mask, $x^*$ is the informative component, and $\Delta x$ is the noise component. $\odot$ is a generalized Hadamard product operation performed between two vectors(matrices). When the inputs both are vectors, denoted as $\boldsymbol{v}$ and $\boldsymbol{m}$, the expression $\boldsymbol{v} \odot \boldsymbol{m}$ refers to the element-wise product. In the case where the first operand is a vector $\boldsymbol{v} \in \mathbb{R}^N$ and the second operand is a matrix $\boldsymbol{M} \in \mathbb{R}^{N \times M}$, the procedure involves duplicating $\boldsymbol{v}$ to form a matrix $\boldsymbol{V} \in \mathbb{R}^{N \times M}$ and then

---

[1]A more formal notation is $\boldsymbol{h}^{(x)}$ as $\boldsymbol{h}$ is dependent on $x$. In this section, for ease of notation, we use $\boldsymbol{h}$ to denote $\boldsymbol{h}^{(x)}$ in contexts where $x$ is fixed or evident from the surrounding discussion.

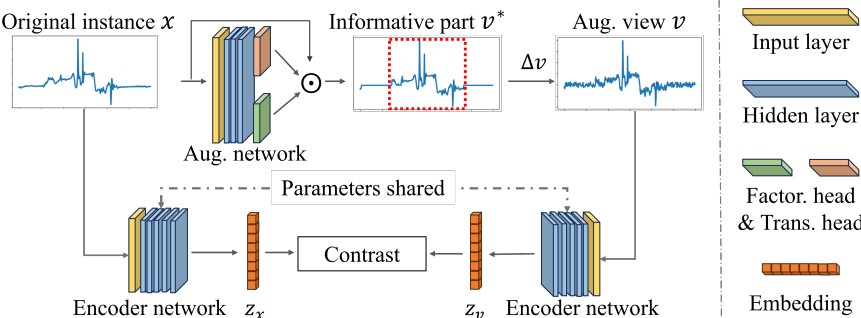

Figure 1: The framework of our AutoTCL. The augmentation network extracts the informative part from the original instance and losslessly transforms it to $v^*$. The encoder network is optimized with the contrastive objective.

applying the standard Hadamard product to $V$ and $M$. If both inputs are matrices of the same shape, the symbol $\odot$ signifies the standard Hadamard product. For the invertible transformation function applied on $x^*$, we present a mask-based instantiation. More sophisticated mechanisms, such as normalizing flows (Kobyzev et al., 2020; Wang et al., 2023), can also be used as plug-and-play components. Specifically, we introduce a non-zero mask $\boldsymbol{g} \in \mathbb{R}^T_{\neq 0}$ to form the transformation, such that $v^* = \boldsymbol{g} \odot x^*$. It is easy to show that such a non-zero mask transformation is lossless as the original $x^*$ can be restored by $x^* = \frac{1}{\boldsymbol{g}} \odot v^*$. Due to the varied nature of time series instances, which might differ even within a dataset, it is impractical to have a universal mask that applies to all instances. For instance, the cutout and jitter transform might not work well for the same time series data. To ensure each instance has a suitable transform adaptively, a parametric mask generator, denoted by $g : \mathbb{R}^{T \times F} \to \mathbb{R}^T_{\neq 0}$, is proposed to generate the non-zero mask by learning for lossless transformation through a data-driven approach. Formally, $\boldsymbol{g} = g(x)$. Then a good view $v$ for contrastive learning can be represented as follows by integrating the factorization function, the mask generator, and introducing random noise for perturbation ($\Delta v$).

$$v = \eta(v^*, \Delta v) = \eta(\boldsymbol{g} \odot x^*, \Delta v)$$
$$= \eta(g(x) \odot h(x) \odot x, \Delta v). \tag{3}$$

**Relationship to existing augmentations for time series.** Various types of data augmentation techniques have been applied to enhance the performance of deep learning on time series data (Wen et al., 2021; Yue et al., 2022; Fan et al., 2020), including time domain, frequency domain, and their hybrids. Our instantiation can be considered a general augmentation framework in the time domain. Most existing augmentation operations in this category, such as cropping, flipping, scaling, and jittering can be unified in our framework. For example, by setting $\eta(v^*, \Delta v) = v^*$, cropping, which deletes a subsequence, can be achieved by letting $\boldsymbol{g} = \mathbb{1}$ and the cropping time steps in $\boldsymbol{h}$ be 0; scaling, which multiplies the input time series by a scaling factor, either a constant or being sampled from a Gaussian distribution, can also be obtained by setting $\boldsymbol{h} = \mathbb{1}$ and $\boldsymbol{g}$ being the scaling factor.

**Practical instantiation with augmentation neural network.** According to the Universal Approximation Theorem (Chapter 6 in (Goodfellow et al., 2016)), we implement $g(x)$ and $h(x)$ with neural networks, respectively. We first utilize the same input layer and a stacked dilated CNN module (Franceschi et al., 2019; Yue et al., 2022) for both $g(x)$ and $h(x)$, respectively. Then, we include two projector heads, a factorization head for $h(x)$, and an transformation head for $g(x)$. The architecture of the overall augmentation network is shown in Fig. 1. To ensure the binary output of the factorization function $h(x)$, we introduce a stochastic mechanism following the factorization head. Specifically, we assume that each element in the output $\boldsymbol{h}$, denoted by $h_i$, is drawn from a Bernoulli distribution parameterized by $\pi_i$, which is calculated by the factorization head.

To enable efficient optimization with gradient-based methods, we approximate the discrete Bernoulli processes with hard binary concrete distributions (Louizos et al., 2018). Specifically, we first draw $\tilde{h}_i$ from a binary concrete distribution with $\pi_i$ indicating the location (Maddison et al., 2017; Jang et al.,

2017). Formally,

$$\tilde{h}_i = \sigma((\log \epsilon - \log(1 - \epsilon) + \log \frac{\pi_i}{1 - \pi_i})/\tau), \tag{4}$$

where $\epsilon \sim \text{Uniform}(0, 1)$ is an independent variable, $\sigma(\cdot)$ is the sigmoid function, and $\tau$ is the temperature. The output of the binary concrete distribution is in the range of (0,1). To further facilitate the binary selection, we stretch $\tilde{h}_i$ the range to $(\gamma, \zeta)$, with $\gamma < 0$ and $\zeta > 1$. Then, the final masking element $h_i$ is obtained by clipping values to the range $[0, 1]$. Formally,

$$h_i = \min(1, \max(\tilde{h}_i(\zeta - \gamma) + \gamma, 0)). \tag{5}$$

$\eta(v^*, \Delta v)$ **as random timestamp masking.** To increase the variance of augmented views, inspired by Dropout (Srivastava et al., 2014) and TS2Vec (Yue et al., 2022), we implement the function $\eta(v^*, \Delta v)$ by randomly masking the hidden representation. Specifically, given a latent vector of a view $v$, after the first hidden layer, we randomly mask it along the time dimension with a binary vector. Each element is sampled independently from a Bernoulli distribution, $\text{Bern}(0.5)$.

## 3.4 TRAINING ALGORITHM

There are two parametric neural networks to be optimized in the proposed framework, i.e., the encoder and the augmentation networks. The augmentation network aims to generate good views with high diversities, and the encoder is trained with a contrastive learning objective. Our method is used as plug and play component and can be used in a wide range of contrastive learning frameworks, such as TS2Vec (Yue et al., 2022) and CoST (Woo et al., 2022). In this section, we first introduce a new objective to train the augmentation network followed by the alternating training of encoder and augmentation networks.

**Training the augmentation network with the principle of relevant information.** Existing optimization techniques for the learnable augmentation and the encoder generally follow the principle of Information Bottleneck (IB) (Tishby et al., 2000), which aims to achieve sufficient and minimal representations (Tian et al., 2020; Yin et al., 2022; Suresh et al., 2021). However, IB relies on the class labels from the downstream task, making it unsuitable for self-supervised training where there are few or no labels. Instead of following previous works that adversarially train the encoder and augmentation networks, which may fail to preserve the semantics in time series data, we train the augmentation network based on the Principle of Relevant Information (PRI) (Principe, 2010). Unlike supervised IB which relies on another variable as well as their joint distributions, PRI only exploits the self-organization of a random variable, making it fully unsupervised. Specifically, with PRI training the augmentation network, we aim to achieve a reduced statistical representation $v^*$ by minimizing the following function.

$$\beta H(v^*) + D(P_x || P_{v^*}), \tag{6}$$

where $\beta$ is the trade-off hyper-parameter, $H(v^*)$ is the entropy of representation variable $v^*$, and $D(P_x || P_{v^*})$ is the divergence between distributions of the original variable $x$ and transformed variable $v^*$. The minimization of $H(v^*)$ aims to reduce uncertainty and obtain statistical regularity in $v^*$ and the second term is for preserving the descriptive power of $v^*$ about $x$.

Given an instance $x$, the transformed informative part $v^*$ is obtained by applying a binary factorization mask $\boldsymbol{h} \in \{0, 1\}^T$ on $x$ and then an invertible transformation function, thus, minimizing the first term in Eq. (6), $H(v^*)$, can be achieved by minimizing the number of non-zero elements in the factorization mask, i.e. $||\boldsymbol{h}||_0$. According to the calculation of $\boldsymbol{h}$ in Eq. (5), we have

$$||\boldsymbol{h}||_0 = \sum_{t=1}^{T} \left( 1 - \sigma(\tau \log \frac{-\gamma(1 - \pi_t)}{\zeta \pi_t}) \right). \tag{7}$$

To preserve the descriptive power of $v^*$ about $x$, we follow existing works and estimate the second term $D(P_x || P_{v^*})$ with the maximum mean discrepancy, $\text{MMD}(P_x, P_{v^*})$ (Gretton et al., 2012). In practice, given a mini-batch of samples, $\mathbb{X}$, and its associated view set $\mathbb{V}^*$, we first compute the embeddings by passing all instances from $\mathbb{X}$ through the function $f$. The same procedure applies to $\mathbb{V}^*$. Then, the loss function to train the augmentation network is shown as follows,

$$\mathcal{L}_{\text{PRI}} = \frac{1}{|\mathbb{X}|} \sum_{x \in \mathbb{X}} \beta ||\boldsymbol{h}^{(x)}||_0 + ||\frac{1}{|\mathbb{X}|} \sum_{x \in \mathbb{X}} f(x) - \frac{1}{|\mathbb{V}^*|} \sum_{v^* \in \mathbb{V}^*} f(v^*)||^2. \tag{8}$$

**Regularization of temporal consistency.** As shown in previous studies (Luo et al., 2023), informative signals tend to be continuous. Thus, we include regularization of temporal consistency when generating the factorization mask. Specifically, given a batch $\mathbb{X}$, for each instance $x \in \mathbb{X}$, we randomly select a time point $a$ as the anchor. Then we randomly select a time point $p$ from the left or right position of $a$ to create a positive pair $(a, p)$. Their mask values $h_a^{(x)}$ and $h_p^{(x)}$ should be similar, compared to another point $n$ that is far away from $a$, whose mask value is denoted by $h_n^{(x)}$. Formally, we have the following triplet loss.

$$\mathcal{L}_t = \frac{1}{|\mathbb{X}|} \sum_{x \in \mathbb{X}} (|h_a^{(x)} - h_p^{(x)}| - |h_a^{(x)} - h_n^{(x)}|). \tag{9}$$

With a trade-off parameter $\lambda$, the final augmentation network loss could be formulated as:

$$\mathcal{L}_{\text{aug}} = \mathcal{L}_{\text{PRI}} + \lambda \mathcal{L}_t \tag{10}$$

**Alternative Training.** To train encoder and augmentation networks, we follow GAN (Goodfellow et al., 2020) to use an alternating training schedule that trains the encoder network $M$ times and then trains the augmentation network one time. $M$ is a hyper-parameter determined by grid search.

## 4 EXPERIMENTS

We compare AutoTCLwith extensive baselines on both forecasting and classification tasks. We also conduct ablation studies to show insights into each component in AutoTCL. Detailed experimental setups, full experimental results, and extensive experiments are presented in the Appendix.

### 4.1 TIME SERIES FORECASTING

**Datasets and baselines.** Six benchmark datasets, ETTh1, ETTh2, ETTm1, (Zhou et al., 2021), Electricity (Dua & Graff, 2017), Weather[2], and Lora dataset are adopted for time series forecasting in both univariate and multivariate settings. Lora dataset is a new introduced real-world dataset that captures the wireless signal data using the LoRa devices[3]. It contains 74 days of data with timestamps. The proposed AutoTCL model is compared to representative state-of-the-art methods such as TS2Vec (Yue et al., 2022), Informer (Zhou et al., 2021), StemGNN (Cao et al., 2020), TCN (Bai et al., 2018), LogTrans (Li et al., 2019), LSTnet (Lai et al., 2018), CoST (Woo et al., 2022), TNC(Tonekaboni et al., 2021), TS-TCC (Eldele et al., 2021), InfoTS (Luo et al., 2023) and N-BEATS (Oreshkin et al., 2020), with N-BEATS being exclusive to univariate forecasting and StemGNN to multivariate.

**Setup.** We follow CoST (Woo et al., 2022) network architecture. A multi-layer dilated CNN module is used for the backbone and we remove the seasonal feature disentangler module. The Augmentation network has the same feature extract architecture and two projectors as shown in Fig. 1. In addition, the proposed AutoTCL is a general contrastive learning framework that can also be combined with more recent methods, such as BTSF (Yang & Hong, 2022), TF-C (Zhang et al., 2022), and LsST (Wang et al., 2022) to further improve accuracy performances. We leave this as our future work. Time series forecasting aims to predict future time stamps, using the last $L_x$ observations. Following the method presented in (Yue et al., 2022), a linear model regularized with L2 norm penalty, is trained to make predictions. Specifically, After pretraining by contrastive learning, the encoder network will be frozen in the following fine-tuning. A linear model is used to map representations to results. The linear model is trained by Linear least squares with l2 regularization in the package sk-learn Pedregosa et al. (2011). We use the default setting during training. This part is kept the same for other competing methods for a fair comparison. In the univariate case, the model's output has a dimension of $L_y$, while in the multivariate case, it has a dimension of $L_y \times F$. The evaluation is based on standard regression metrics, Mean Squared Error (MSE), and Mean Absolute Error (MAE). To comprehensively evaluate the performances, we consider different prediction lengths, $L_y$.

---

[2]https://www.ncei.noaa.gov/data/local-climatological-data/

[3]https://lora-alliance.org/

**Results.** For each dataset, we calculate the average forecasting performances in both univariate and multivariate settings. The results are shown in Table 1 and Table 2, respectively. The detailed results of univariate and multivariate time series forecasting can be found in Table 10 and Table 11 in the Appendix. From these tables, we have several observations. First, in general, contrastive learning methods, including AutoTCL, TS2vec, CoST, and InfoTS, achieve better performances compared to traditional baselines, indicating the effectiveness of contrastive learning for learning time series representations. Second, the consistent improvement of our method over CoST indicates that universal data augmentations may not be the most informative for generating positive pairs in various datasets. Specifically, Compared to CoST, AutoTCL decreases both MAE and MSE in all datasets in the univariate setting. On average, AutoTCL decreases the average MSE by 6.5% and the average MAE by 4.8% in the univariate setting. This is because AutoTCL can adaptively learn the most suitable augmentations in a data-driven manner, preserving semantics and ensuring sufficient variance. Encoders trained with these informative augmentations lead to representations with higher quality. In the multivariate setting, AutoTCL outperforms CoST in 7 cases. On average, it decreases the average MSE by 2.9% and the average MAE by 1.2%.

Table 1: Univariate time series forecasting results.

| | AutoTCL | | TS2Vec | | Informer | | LogTrans | | N-BEATS | | TCN | | CoST | | TNC | | TS-TCC | | InfoTS | |
|---|---|---|---|---|---|---|---|---|---|---|---|---|---|---|---|---|---|---|---|---|
| Dataset | MSE | MAE | MSE | MAE | MSE | MAE | MSE | MAE | MSE | MAE | MSE | MAE | MSE | MAE | MSE | MAE | MSE | MAE | MSE | MAE |
| ETTh$_1$ | **0.076** | **0.207** | 0.110 | 0.252 | 0.186 | 0.347 | 0.196 | 0.365 | 0.218 | 0.375 | 0.263 | 0.431 | 0.091 | 0.228 | 0.150 | 0.303 | 0.168 | 0.316 | 0.091 | 0.227 |
| ETTh$_2$ | 0.158 | **0.299** | 0.170 | 0.321 | 0.204 | 0.358 | 0.217 | 0.391 | 0.326 | 0.442 | 0.219 | 0.362 | 0.161 | 0.307 | 0.168 | 0.322 | 0.298 | 0.428 | **0.149** | **0.299** |
| ETTm$_1$ | 0.046 | 0.154 | 0.069 | 0.186 | 0.241 | 0.382 | 0.270 | 0.416 | 0.162 | 0.326 | 0.200 | 0.349 | 0.054 | 0.164 | 0.069 | 0.191 | 0.158 | 0.299 | 0.050 | 0.157 |
| Elec. | **0.366** | **0.345** | 0.393 | 0.370 | 0.464 | 0.388 | 0.744 | 0.528 | 0.727 | 0.482 | 0.525 | 0.423 | 0.375 | 0.353 | 0.378 | 0.359 | 0.511 | 0.603 | 0.369 | 0.348 |
| WTH | 0.160 | **0.287** | 0.181 | 0.308 | 0.241 | 0.370 | 0.280 | 0.411 | 0.256 | 0.374 | 0.166 | 0.291 | 0.183 | 0.307 | 0.175 | 0.303 | 0.302 | 0.442 | 0.176 | 0.304 |
| Lora | **0.177** | **0.273** | 0.356 | 0.385 | 1.574 | 0.999 | 0.656 | 0.550 | 0.311 | 0.349 | 1.160 | 0.927 | 0.186 | 0.282 | 0.620 | 0.565 | 0.490 | 0.591 | 0.333 | 0.325 |
| Avg. | **0.157** | **0.258** | 0.207 | 0.301 | 0.486 | 0.477 | 0.382 | 0.441 | 0.320 | 0.388 | 0.419 | 0.465 | 0.168 | 0.271 | 0.256 | 0.340 | 0.315 | 0.441 | 0.188 | 0.274 |

Table 2: Multivariate time series forecasting results.

| | AutoTCL | | TS2Vec | | Informer | | LogTrans | | StemGNN | | TCN | | CoST | | TNC | | TS-TCC | | InfoTS | |
|---|---|---|---|---|---|---|---|---|---|---|---|---|---|---|---|---|---|---|---|---|
| Dataset | MSE | MAE | MSE | MAE | MSE | MAE | MSE | MAE | MSE | MAE | MSE | MAE | MSE | MAE | MSE | MAE | MSE | MAE | MSE | MAE |
| ETTh$_1$ | 0.656 | 0.590 | 0.788 | 0.646 | 0.907 | 0.739 | 1.043 | 0.890 | 0.738 | 0.632 | 1.021 | 0.816 | **0.650** | **0.585** | 0.904 | 0.702 | 0.748 | 0.635 | 0.784 | 1.622 |
| ETTh$_2$ | **1.191** | **0.815** | 1.566 | 0.937 | 2.371 | 1.199 | 2.898 | 1.356 | 1.940 | 1.077 | 2.574 | 1.265 | 1.283 | 0.851 | 1.869 | 1.053 | 2.120 | 1.109 | 1.474 | 0.914 |
| ETTm$_1$ | **0.409** | 0.441 | 0.628 | 0.553 | 0.749 | 0.640 | 0.965 | 0.914 | 0.729 | 0.626 | 0.818 | 0.849 | 0.409 | 0.439 | 0.740 | 0.599 | 0.612 | 0.564 | 0.568 | 0.521 |
| Elec. | 0.175 | 0.272 | 0.319 | 0.397 | 0.495 | 0.488 | 0.351 | 0.412 | 0.501 | 0.489 | 0.332 | 0.404 | **0.165** | **0.268** | 0.387 | 0.446 | 0.511 | 0.602 | 0.289 | 0.376 |
| WTH | 0.423 | **0.457** | 0.451 | 0.474 | 0.574 | 0.552 | 0.645 | 0.617 | **0.353** | 0.593 | 0.440 | 0.461 | 0.430 | 0.464 | 0.441 | 0.466 | 0.483 | 0.535 | 0.455 | 0.472 |
| Lora | 0.346 | 0.372 | 0.356 | 0.384 | 0.743 | 0.586 | 0.766 | 0.520 | **0.258** | 0.492 | 1.013 | 0.814 | 0.350 | 0.378 | 0.590 | 0.518 | 0.490 | 0.591 | 0.345 | **0.368** |
| Avg. | **0.545** | **0.499** | 0.697 | 0.571 | 0.990 | 0.708 | 1.138 | 0.798 | 0.753 | 0.651 | 1.057 | 0.781 | 0.561 | 0.505 | 0.837 | 0.637 | 0.838 | 0.675 | 0.665 | 0.556 |

## 4.2 TIME SERIES CLASSIFICATION

**Datasets and baselines.** For the classification task, we evaluate our method on the UEA dataset (Dau et al., 2019), which contains 30 multivariate time series datasets. We compare our method with 8 state-of-the-art baselines, including TS2Vec (Yue et al., 2022), T-Loss (Franceschi et al., 2019), TNC (Tonekaboni et al., 2021), TS-TCC (Eldele et al., 2021), TST (Zerveas et al., 2021), DTW (Chen et al., 2013), TF-C (Zhang et al., 2022) and InfoTS (Luo et al., 2023).

**Setup.** We use TS2Vec (Yue et al., 2022) network architecture. In the training stage, we use the same strategy as the forecasting tasks which could be found in Appendix. We follow the previous setting (Yue et al., 2022) that the evaluation is conducted in a standard supervised manner. A radial basis function kernel SVM classifier is trained on the training set and then makes predictions on test data. We report two metrics in the results, accuracy(ACC) and rank(RANK).

**Results.** The results on the 30 UEA datasets are summarized in Table 3. The detailed results can be found in Table 12 in the Appendix. Overall, AutoTCL substantially outperforms baselines with an average rank value 2.3. As shown in Table 12, our method achieves the best results in 16 out of 30 datasets. In addition, it improves the classification accuracy by $1.6\%$, on average, over the second-best baseline, InfoTS. The comprehensive comparison indicates the effectiveness of the proposed method.

## 4.3 ABLATION STUDY AND MODEL ANALYSIS.

In this set of experiments, we conduct ablation studies to investigate the effectiveness of each component in the proposed method. To present deep insights into automatic data augmentation and

Table 3: Classification result of the UEA dataset

| Dataset | AutoTCL | TS2Vec | T-Loss | TNC | TS-TCC | TST | DTW | TF-C | InfoTS |
|---------|---------|--------|--------|-----|--------|-----|-----|------|--------|
| Avg. ACC | **0.742** | 0.704 | 0.658 | 0.670 | 0.668 | 0.617 | 0.629 | 0.298 | 0.730 |
| Avg. RANK | **2.300** | 3.700 | 4.667 | 5.433 | 5.133 | 6.133 | 5.400 | 8.200 | 2.367 |

factorization, we compare AutoTCL with multiple groups of variants. (1)**W/o** $h(x)$, **W/o** $g(x)$, and **W/o** $\Delta v$ are ablation studies about the effectiveness of each part of AutoTCL. In our experiments, **W/o** $h(x)$ means the whole input instance would be regarded as the informative part. **W/o** $g(x)$ represents the transformation head $g(x)$ would be replaced by all 1 vectors and no noise will be added in **W/o** $\Delta v$ setting. (2) **Cutout** and **Jitter** are two commonly adopted data augmentation techniques for time series contrastive learning. We replace the augmentation network in AutoTCL with these two static transformations as variants. (3) **Adversarial** training is routinely adopted in the literature to learn views for contrastive learning. For this variant, we adversarially train the augmentation network by minimizing the mutual information between views and original instances, approximated by the InfoNCE (Tian et al., 2020). (4), **Random Aug.** randomly select augmentation operations from **Cutout** and **Jitter** with different parameters. The parameter of cutout ranges from $0.3$ to $0.8$. The mean of **Jitter** is set to 0, and the standard deviation ranges from $0.3$ to $1.0$. We report the averaged performances in Table 4 and the full results are shown in Table 6 in Appendix.

We have several observations in Table 4. First, by removing the factorization head, **W/o** $h(x)$ increase the MSE by 10.19% and MAE by 4.65% respectively, verifying the effectiveness of the factorization. The comparison between AutoTCLand **W/o** $g(x)$, indicates the importance of invertible view generation. Specifically, **W/o** $g(x)$ increases the MSE by 37.6% and MAE by 9.3%; The difference between AutoTCLand **W/o** $\Delta v$ indicates the importance of diversity in data augmentation. Second, the comparison between **W/o Aug** and **Cutout** shows that universal and non-parametric augmentation techniques may harm the performances of time series contrastive learning. On average, **Cutout** performs even worse than **W/o Aug**. This observation is consistent with the conclusion drawn in TS2Vec (Yue et al., 2022). By adaptive learning suitable augmentations, our methods can consistently and significantly outperform these baselines. Third, with the augmentation network trained in an adversarial manner, the variant, **Adversarial** improves the performances, indicating the necessity of adaptive augmentation for time series data. However, overlooking the semantic preservation may generate trivial augmented views, hindering the performance of downstream contrastive learning. On the other hand, our method achieves the best performances in most cases, especially for forecasting long periods, which verifies the advantage of our training algorithm. The comparison between AutoTCL and **Random Aug.** further indicates the advantage of parametric augmentation.

Table 4: Ablation studies and model analysis

| Dataset | AutoTCL | | W/o $h(x)$ | | W/o $g(x)$ | | W/o $\Delta v$ | | W/o Aug | | Cutout | | Jitter | | Adversarial | | Random Aug. | |
|---------|-----|-----|-----|-----|-----|-----|-----|-----|-----|-----|-----|-----|-----|-----|-----|-----|-----|-----|
| | MSE | MAE | MSE | MAE | MSE | MAE | MSE | MAE | MSE | MAE | MSE | MAE | MSE | MAE | MSE | MAE | MSE | MAE |
| ETTh$_1$ | **0.076** | **0.207** | 0.077 | 0.208 | 0.078 | 0.209 | 0.086 | 0.219 | 0.095 | 0.231 | 0.088 | 0.221 | 0.086 | 0.219 | 0.089 | 0.224 | 0.112 | 0.254 |
| ETTh$_2$ | **0.158** | **0.299** | 0.168 | 0.305 | 0.178 | 0.312 | 0.176 | 0.311 | 0.170 | 0.309 | 0.160 | 0.306 | 0.173 | 0.317 | 0.187 | 0.319 | 0.168 | 0.321 |
| ETTm$_1$ | **0.046** | **0.154** | 0.052 | 0.161 | 0.050 | 0.159 | 0.051 | 0.163 | 0.053 | 0.162 | 0.053 | 0.164 | 0.056 | 0.170 | 0.052 | 0.163 | 0.065 | 0.187 |
| Elec. | **0.365** | 0.348 | 0.371 | 0.349 | **0.365** | 0.348 | 0.366 | 0.347 | 0.368 | 0.354 | 0.367 | **0.345** | 0.366 | 0.344 | **0.365** | **0.345** | 0.376 | 0.358 |
| WTH | **0.160** | **0.287** | 0.172 | 0.301 | 0.166 | 0.295 | 0.164 | 0.293 | 0.183 | 0.309 | 0.167 | 0.294 | 0.174 | 0.304 | 0.166 | 0.294 | 0.184 | 0.310 |
| Lora | **0.177** | **0.273** | 0.237 | 0.309 | 0.489 | 0.385 | 0.304 | 0.361 | 0.711 | 0.412 | 0.783 | 0.442 | 0.285 | 0.346 | 0.445 | 0.373 | 0.191 | 0.299 |
| Avg. | **0.157** | **0.258** | 0.173 | 0.270 | 0.216 | 0.282 | 0.185 | 0.280 | 0.260 | 0.294 | 0.266 | 0.394 | 0.184 | 0.281 | 0.212 | 0.284 | 0.176 | 0.286 |

## 5 CONCLUSION AND FUTURE WORK

We present a novel factorization-based augmentation framework for time series representation learning in the self-supervised setting. Theoretical analysis from the information theory perspective shows that the proposed framework is more flexible to persevere semantics and includes sufficient variances to augment views. On top of that, we provide a simple and effective instantiation and an efficient training algorithm. With time series forecasting as the downstream task, we compare the proposed method, AutoTCL, with representative methods and verify its effectiveness. In addition, AutoTCL exploits the informative part of time series data, which might help users better understand the time series data. In the future, we plan to investigate the usage of parametric augmentation in other contrastive learning frameworks and extend the technique to other domains.

## ACKNOWLEDGMENTS

This project was partially supported by NSF grants IIS-2331908 and CNS-2150010. The views and conclusions contained in this paper are those of the authors and should not be interpreted as representing any funding agencies.

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

# Appendix: Parametric Augmentation for Time Series Contrastive Learning

## A  Notations

Important notations are summarized in Table 5.

Table 5: Notations and their meanings.

| Symbol | Meaning |
|---|---|
| $x, \mathsf{x}$ | Time series |
| $x^*, \mathsf{x}^*$ | Informative part of $x$ |
| $\Delta x, \Delta \mathsf{x}$ | Task-irrelevant part of $x$ |
| $v, \mathsf{v}$ | Augmented view |
| $v^*, \mathsf{v}^*$ | Informative part of $v$ |
| $\Delta v, \Delta \mathsf{v}$ | Task-irrelevant noise |
| $y, \mathsf{y}$ | Downstream task label |
| $T$ | Length of time series |
| $F$ | Number of features of time series |
| $D$ | Dimensions of hidden representations |
| $f$ | Encoder function |
| $g$ | Transformation function |
| $h$ | Factorization function |
| $\eta$ | Augmentation function |
| $H(\cdot)$ | Entropy |
| $MI(\cdot; \cdot)$ | Mutual information |
| $\boldsymbol{h}$ | Factorization mask |
| $\boldsymbol{g}$ | Transformation mask |
| $h_i$ | $i$-th element of $\boldsymbol{h}$ |
| $\tilde{h}_i$ | Intermediate result from concrete distribution |
| $\pi_i$ | Parameter in Bern. distribution |
| $\epsilon$ | Random variable from uniform distribution |
| $\tau$ | Temperature in Concrete distribution |
| $\zeta, \gamma$ | Hyper-parameters in hard concrete distribution |
| $\mathbb{X}$ | A batch/set of time series instances |
| $a, p, n$ | Anchor, positive and negative time stamp |
| $\beta, \lambda$ | Trade-off hyper-parameters |
| $M$ | Hyper-parameter for training |

## B  Detailed proofs

**Property 1.** *If view $\mathsf{v}$ is generated from $\mathsf{x}$ with an invertible function $\mathsf{v} = g(\mathsf{x})$. Then $H(\mathsf{v}) = H(\mathsf{x}) = MI(\mathsf{x}; \mathsf{v})$, where $H(\mathsf{x}), H(\mathsf{v})$ are entropy of variables $\mathsf{x}$ and $\mathsf{v}$, respectively; $MI(\mathsf{v}; \mathsf{x})$ is the mutual information between $\mathsf{v}$ and $\mathsf{x}$.*

*Proof.* Since $g$ is an invertible function and $\mathsf{v} = g(\mathsf{x})$, we have an one-to-one mapping between variables $\mathsf{v}$ and $\mathsf{x}$. Thus, $v = g(x)$ for each pair of $x$ and $v$. We have $\mathbb{P}[\mathsf{x} = x] = \mathbb{P}[\mathsf{v} = v]$. From the definition of Shannon entropy, we have

$$
\begin{aligned}
H(\mathsf{x}) &= -\sum_x p(x) \log p(x) = -\sum_x \mathbb{P}[\mathsf{x} = x] \log \mathbb{P}[\mathsf{x} = x] \\
&= -\sum_v \mathbb{P}[\mathsf{v} = v] \log \mathbb{P}[\mathsf{v} = v] = -\sum_x p(v) \log p(v) \\
&= H(\mathsf{v}).
\end{aligned}
$$

From the definition of conditional entropy, we have

$$H(\mathsf{x}|\mathsf{v}) = \sum_{v,x} p(v,x) \log \frac{p(v,x)}{p(v)},$$

$$H(\mathsf{x}|\mathsf{v}) = \sum_{v,x} p(v) \log \frac{p(v)}{p(v)} = 0.$$

The above results in the mutual information between $\mathsf{v}$ and $\mathsf{x}$, given by

$$\mathrm{MI}(\mathsf{v};\mathsf{x}) = H(\mathsf{x}) - H(\mathsf{x}|\mathsf{v}) = H(\mathsf{v}).$$

$\square$

**Property 2** *(Task agnostic label preserving). If a variable $\mathsf{v}$ is a good view of $\mathsf{x}$, and the downstream task label $\mathsf{y}$ (although not visible to training) is independent to noise in $\mathsf{x}$, the mutual information between $\mathsf{v}$ and $\mathsf{y}$ is equivalent to that between raw input $\mathsf{x}$ and $\mathsf{y}$, i.e., $MI(\mathsf{v};\mathsf{y}) = MI(\mathsf{x};\mathsf{y})$.*

*Proof.* From the definition of the good view, we have

$$\mathsf{x} = \mathsf{x}^* + \Delta\mathsf{x}$$
$$\mathsf{v} = \eta(g(\mathsf{x}^*), \Delta\mathsf{v}).$$

We first analyze the relationship between $\mathrm{MI}(\mathsf{x},\mathsf{y})$ and $\mathrm{MI}(\mathsf{x}^*,\mathsf{y})$.

$$\mathrm{MI}(\mathsf{x},\mathsf{y}) = H(\mathsf{y}) - H(\mathsf{y}|\mathsf{x})$$

$$= H(\mathsf{y}) + \sum_{x,y} p(x,y) \log \frac{p(x,y)}{p(x)}$$

$$= H(\mathsf{y}) + \sum_{x^*,\Delta x,y} p(x^*,\Delta x,y) \log \frac{p(x^*,\Delta x,y)}{p(x^*,\Delta x)}$$

$$= H(\mathsf{y}) + \sum_{x^*,\Delta x,y} p(x^*,\Delta x,y) \log \frac{p(\Delta x,y|x^*)}{p(\Delta x|x^*)}.$$

With the safe independence assumption, we have

$$p(\Delta x, y|x^*) = p(\Delta x|x^*)p(y|x^*).$$

Thus, we show that

$$\mathrm{MI}(\mathsf{x},\mathsf{y}) = H(\mathsf{y}) + \sum_{x^*,\Delta x,y} p(x^*,\Delta x,y) \log \frac{p(x^*,y)}{p(x^*)}$$

$$= H(\mathsf{y}) + \sum_{x^*,y} p(x^*,y) \log \frac{p(x^*,y)}{p(x^*)}$$

$$= H(\mathsf{y}) - H(\mathsf{y}|\mathsf{x}^*)$$

$$= \mathrm{MI}(\mathsf{x}^*,\mathsf{y}).$$

Letting $\mathsf{v}^* = g(\mathsf{x}^*)$, from the Property 1, we have

$$\mathrm{MI}(\mathsf{x}^*,\mathsf{y}) = \mathrm{MI}(\mathsf{v}^*,\mathsf{y}).$$

Since $\Delta\mathsf{v}$ is a random noise and is independent to the label $\mathsf{y}$, similarly, we have

$$\mathrm{MI}(\mathsf{v},\mathsf{y}) = \mathrm{MI}(\eta(\mathsf{v}^*,\Delta v),\mathsf{y})$$
$$= \mathrm{MI}((\mathsf{v}^*,\Delta v),\mathsf{y})$$
$$= \mathrm{MI}(\mathsf{v}^*,\mathsf{y})$$

Combining them together results in

$$\mathrm{MI}(\mathsf{v},\mathsf{y}) = \mathrm{MI}(\mathsf{x},\mathsf{y}).$$

$\square$

**Property 3.** *(Containing more information). A good view* v *contains more information comparing to the raw input* x, *i.e.,* $H(v) \geq H(x)$.

*Proof.* Since $\Delta v$ denotes the included random noise, we assume that its generation is independent of the augmented view $v^*$. Thus we have

$$\text{MI}(\Delta v, v^*) = 0. \tag{11}$$

Further, with our decomposing model, we can rewrite the entropy of x as the joint entropy of $x^*$ and $\Delta x$. Formally, we have

$$H(x) = H(x^*, \Delta x) = H(x^*) + H(\Delta x) - \text{MI}(\Delta x, x^*).$$

Then $H(x^*) = H(v^*)$ holds (Property 1). From the definition of the good view, we have $H(\Delta v) \geq H(\Delta x)$. Thus, we have

$$
\begin{aligned}
H(x) &= H(x^*) + H(\Delta x) - \text{MI}(\Delta x, x^*) \\
&\leq H(v^*) + H(\Delta v) \\
&= H(v^*) + H(\Delta v) - \text{MI}(\Delta v, v^*) \\
&= H(\Delta v, v^*) = H(\eta(v^*, \Delta v)) \\
&= H(v).
\end{aligned}
$$

$\square$

**Derivation of Eq. (7)** As described in Section 3.4, the factorization mask $h$ is generated with a hard concrete distribution. Thus, the number of non-zero entries in $h$ can be reformulated with

$$||\boldsymbol{h}||_0 = \sum_t (1 - \mathbb{P}_{h_t}(0)),$$

where $\mathbb{P}_{h_t}(0)$ is the cumulative distribution function (CDF) of $h_t$ (before clipping). We let $\text{S}(\cdot)$ be an affine function of the stretch process in Eq. (5), such that

$$h_t = \text{S}(\tilde{h}_t) = \tilde{h}_t(\zeta - \gamma) + \gamma,$$

where $\gamma \in (-\infty, 0)$ and $\zeta \in (1, \infty)$. As derived in (Maddison et al., 2017), the density of $h_t$ is

$$p_{\mathsf{h}_t}(h_t) = \frac{\tau \alpha_t h_t^{-\tau-1}(1 - h_t)^{-\tau-1}}{(\alpha_t h_t^{-\tau} + (1 - h_t)^{-\tau})^2},$$

where $\alpha_t = \log \frac{\pi_t}{1-\pi_t}$. The CDF of $h_t$ reads

$$\mathbb{P}_{\mathsf{h}_t}(h_t) = \sigma((\log h_t - \log(1 - h_t))\tau - \alpha_t).$$

Thus, the probability density function of $h_t$ is

$$
\begin{aligned}
p_{\mathsf{h}_t}(h_t) &= p_{\tilde{\mathsf{h}}_t}(\text{S}^{-1}(h_t)) \left| \frac{\partial}{\partial h_t} \text{S}^{-1}(h_t) \right| \\
&= \frac{(\zeta - \gamma)\tau \alpha_t (h_t - \gamma)^{-\tau-1}(\zeta - h_t)^{-\tau-1}}{(\alpha_t (h_t - \gamma)^{-\tau} + (\zeta - h_t)^{-\tau})^2}.
\end{aligned}
$$

The CDF of $h_t$ is given by

$$
\begin{aligned}
\mathbb{P}_{\mathsf{h}_t}(h_t) &= \mathbb{P}_{\tilde{\mathsf{h}}_t}(\text{S}^{-1}(h_t)) \\
&= \sigma((\log(h_t - \gamma) - \log(\zeta - h_t))\tau - \alpha_t).
\end{aligned}
$$

When setting $h_t = 0$, we have the

$$\mathbb{P}_{\mathsf{h}_t}(0) = \sigma(\tau \log \frac{-\gamma}{\zeta} - \alpha_t).$$

## C  Implementation details

### C.1  Training the encoder with local and global contrasts.

Similar to the augmentation network, our method can work with different architectures. We formulate the feature extraction encoder as $f : \mathbb{R}^{T \times F} \to \mathbb{R}^D$, where $D$ is the dimensionality of output embeddings. Following existing work (Luo et al., 2023), we use both global and local contrastive losses.

**Global contrast** aims to improve the inter-instance robustness for representation learning. Given a batch of time series instances $\mathbb{X}$, for each instance $x \in \mathbb{X}$, we generate an augmented view $v$. Such a pair of the original instance $x$ and the corresponding view $v$ is then used as a positive pair. Other pairs of instances and views are treated as negative pairs. Formally, $(x, v')$ is a negative pair, where $v'$ is an augmented view of $x'$ and $x' \neq x$. Following (Chen et al., 2020), we use the InfoNCE as the global-wise contrastive loss to train the encoder network. Formally, we have

$$\mathcal{L}_g = -\frac{1}{|\mathbb{X}|} \sum_{x \in \mathbb{X}} \log \frac{\exp(\mathrm{sim}(\boldsymbol{z}_x, \boldsymbol{z}_v))}{\sum_{x' \in \mathbb{X}_B} \exp(\mathrm{sim}(\boldsymbol{z}_x, \boldsymbol{z}_{v'}))}, \tag{12}$$

where $\boldsymbol{z}_x = f(x), \boldsymbol{z}_v = f(v), \boldsymbol{z}_{x'} = f(x'), \boldsymbol{z}_{v'} = f(v')$ are representations of $x, v, x'$ and $v'$, respectively.

**Local contrast** is designed to enhance the encoder network to capture the intra-instance relationship. Given an augmented view $v$, we first segment it into a set of subsequences $\mathbb{S}$, where each subsequence $s \in \mathbb{S}$ has length $L$. Following (Tonekaboni et al., 2021), two close subsequences $(s, p)$ are considered as a positive pair, and the ones with a large distance lead to a negative pair. Formally, the loss of local contrast is:

$$\mathcal{L}_{l_x} = -\frac{1}{|\mathbb{S}|} \sum_{s \in \mathbb{S}} \log \frac{\exp(\mathrm{sim}(\boldsymbol{z}_s, \boldsymbol{z}_p))}{\exp(\mathrm{sim}(\boldsymbol{z}_s, \boldsymbol{z}_p)) + \sum_{j \in \bar{\mathcal{N}}_s} \exp(\mathrm{sim}(\boldsymbol{z}_s, \boldsymbol{z}_j))}, \tag{13}$$

where $\bar{\mathcal{N}}_s$ is the set of negative pairs for a subsequence $s$. $\boldsymbol{z}_s = f(s), \boldsymbol{z}_p = f(p), \boldsymbol{z}_n = f(n)$ are representations of $s, p$ and $n$ generated by function $f$, respectively. Considering all instances in a batch, we have

$$\mathcal{L}_l = \frac{1}{|\mathbb{X}|} \sum_{x \in \mathbb{X}} \mathcal{L}_{l_x}. \tag{14}$$

With both local and global contrasts, we have our contrastive loss as follows.

$$\mathcal{L}_{\mathrm{con}} = \mathcal{L}_g + \alpha \mathcal{L}_l, \tag{15}$$

where $\alpha$ is the hyper-parameter to achieve the trade-off between global and local losses.

### C.2  Training algorithm

In the training stage, AutoTCL optimizes the augmentation network and encoder network simultaneously. Similar to GAN (Goodfellow et al., 2016), these networks were randomly initialized. Different from GAN, AutoTCL is less affected by the problem of gradient explosion and mode collapse, because our encoder network aims to embed the information part from different views rather than distinguish them. Although our argumentation network tries to reduce the distribution between original instances and arguments, AutoTCL augmentations preserve the information part by using a reversible mapping function, which alleviates the mode collapse problem. Our training algorithm is shown in Alg. 1 .

## D  Experimental settings

All experiments are conducted on a Linux machine with 8 NVIDIA A100 GPUs, each with 40GB of memory. The software environment is CUDA 11.6 and Driver Version 520.61.05. We used Python 3.9.13 and Pytorch 1.12.1 to construct our project.

---

**Algorithm 1** AutoTCL training algorithm

---

**Require:** augmentation network $f_{\text{aug}}$, encoder network $f_{\text{enc}}$, epochs $E$, a hyperparameter $M$,
epoch $\leftarrow 0$
**while** epoch $< E$ **do**
    **for** $x$ in dataset **do**
        $x_a \leftarrow f_{\text{aug}}(x)$
        $\boldsymbol{z}_x \leftarrow f_{\text{enc}}(x)$
        $\boldsymbol{z}_a \leftarrow f_{\text{enc}}(x_a)$
        **if** epoch$\%M == 0$ **then**
            Compute loss with using Eq. (10)
            Update parameters in $f_{\text{aug}}$ with backpropagation
        **end if**
        Compute loss with using Eq. (15)
        Update parameters in $f_{\text{enc}}$ with backpropagation
    **end for**
    epoch $\leftarrow$ epoch $+ 1$
**end while**

---

## D.1 BASELINE SETTINGS

In forecasting tasks, we conducted baseline methods on six benchmark datasets by following the experiment setting of TS2Vec(Yue et al., 2022) for most baseline methods, such as Informer (Zhou et al., 2021), (Tonekaboni et al., 2021), StemGNN (Cao et al., 2020), TCN (Bai et al., 2018), N-BEATS (Oreshkin et al., 2020), etc. For TS2Vec(Yue et al., 2022), CoST (Woo et al., 2022), we followed its code default setting for Lora and Weather datasets. The representation dimension was 320 and the learning rate and batch size were $0.001$ and $8$. For InfoTS (Luo et al., 2023), We used the default setting to conduct experiments. As for TS-TCC (Eldele et al., 2021) in forecasting tasks, we used the Epilepsy config as the default config and modified the network model to make the input and output channels remain the same. Due to its pooling layers, the network would require 3 times the lengths of inputs of other baselines which is unfair for forecasting tasks. In the experiments, we used another interpolate layer to make the length of input data and prediction data the same. In classification tasks, similar to the forecasting task, we followed the experiment setting of TS2Vec(Yue et al., 2022). In TF-C (Zhang et al., 2022) classification experiments, we use its HAR config as the default setting. Similar to TS-TCC, we modifie the network so that the transformer encoder could fit the input length and the pre-train dataset is the same as the fine tune dataset.

## D.2 HYPERPARAMETERS

In our experiments, we used grid search to obtain the best performance. We used the same strategy in forecasting and classification tasks that each dataset had its own group of hyperparameters. We provided all of the hyperparameters as well as their configurations in the following:

- Optimizer: Two Adam optimizers (Kingma & Ba, 2014) were used for the augmentation network and feature extraction network with learning rate and other hyperparameters were setting with default decay rates setting to 0.001 and (0.9,0.999) respectively.

- Encoder architecture: The depth of the multi-layer dilated CNN module and the hidden dimension were designed to be able to change, which were searched in $\{6, 7, 8, 9, 10\}$ and $\{256, 128, 64, 32, 16, 8\}$. In training, we used a designed dropout rate to avoid overfitting, which was tuned in $[0.01, 1]$.

- Augmentation architecture: Same as encoder, the depth of multi-layer dilated CNN module and hidden dimension are hyperparameters, searched in $\{1, 2, 3, 4, 5\}$ and $\{256, 128, 64, 32, 16, 8\}$ and as mention in equation Eq. 5, $\zeta$ is another hyperparameter, tuned in $[0.0005, 0.5]$.

- Trade-off hyperparameters: $\beta$ in Eq. (8), and $\lambda$ in Eq. (10) are tuned in $[0, 0.3]$.

- Alternating training hyperparameters: $M$ in Sec. (3.4) is tuned in $\{1, 2, 4, 8, 16, 32\}$.

### D.3 EXTRA EXPERIMENTS

#### D.3.1 PERFORMANCE WITH DIFFERENT BACKBONES

As a general framework for time series contrastive learning, AutoTCL can be used as a plug-and-play component to boost performance. To further verify the generalization capacity of AutoTCL. Table 6 shows the full results of ablation studies and model analysis with CoST as the backbone. In addition, we adopt Ts2vec (Yue et al., 2022) as the backbone and show the results in Table 7. We can draw similar conclusions that by adaptively selecting the optimal augmentations with the principle of relevant information, AutoTCL can outperform the vanilla TS2vec and other baselines.

Table 6: Ablation studies using CoST backbone

| Dataset | $L_y$ | AutoTCL MSE | AutoTCL MAE | W/o $h(x)$ MSE | W/o $h(x)$ MAE | W/o $g(x)$ MSE | W/o $g(x)$ MAE | W/o $\Delta V$ MSE | W/o $\Delta V$ MAE | W/o Aug MSE | W/o Aug MAE | Cutout MSE | Cutout MAE | Jitter MSE | Jitter MAE | Adversarial MSE | Adversarial MAE | Random Aug. MSE | Random Aug. MAE |
|---|---|---|---|---|---|---|---|---|---|---|---|---|---|---|---|---|---|---|---|
| ETTh$_1$ | 24 | 0.037 | 0.148 | 0.037 | 0.148 | 0.037 | 0.148 | 0.038 | 0.149 | 0.037 | 0.148 | 0.037 | 0.147 | 0.038 | 0.147 | 0.039 | 0.149 | 0.056 | 0.178 |
| | 48 | 0.054 | 0.176 | 0.054 | 0.176 | 0.054 | 0.177 | 0.055 | 0.180 | 0.055 | 0.178 | 0.053 | 0.175 | 0.054 | 0.176 | 0.056 | 0.180 | 0.076 | 0.208 |
| | 168 | 0.078 | 0.210 | 0.079 | 0.210 | 0.080 | 0.211 | 0.083 | 0.217 | 0.100 | 0.237 | 0.078 | 0.210 | 0.081 | 0.212 | 0.090 | 0.227 | 0.132 | 0.278 |
| | 336 | 0.093 | 0.231 | 0.093 | 0.231 | 0.094 | 0.232 | 0.096 | 0.234 | 0.108 | 0.251 | 0.092 | 0.230 | 0.095 | 0.233 | 0.106 | 0.250 | 0.137 | 0.289 |
| | 720 | 0.120 | 0.272 | 0.121 | 0.274 | 0.124 | 0.277 | 0.157 | 0.317 | 0.175 | 0.340 | 0.179 | 0.345 | 0.163 | 0.325 | 0.152 | 0.313 | 0.157 | 0.318 |
| ETTh$_2$ | 24 | 0.079 | 0.206 | 0.077 | 0.204 | 0.076 | 0.205 | 0.079 | 0.209 | 0.078 | 0.208 | 0.076 | 0.204 | 0.092 | 0.217 | 0.078 | 0.207 | 0.090 | 0.229 |
| | 48 | 0.117 | 0.255 | 0.124 | 0.258 | 0.113 | 0.256 | 0.118 | 0.259 | 0.127 | 0.259 | 0.110 | 0.253 | 0.135 | 0.272 | 0.124 | 0.265 | 0.125 | 0.272 |
| | 168 | 0.176 | 0.319 | 0.191 | 0.329 | 0.212 | 0.346 | 0.240 | 0.358 | 0.220 | 0.347 | 0.191 | 0.340 | 0.207 | 0.356 | 0.227 | 0.361 | 0.175 | 0.331 |
| | 336 | 0.193 | 0.344 | 0.201 | 0.350 | 0.243 | 0.371 | 0.204 | 0.349 | 0.200 | 0.357 | 0.201 | 0.355 | 0.212 | 0.366 | 0.253 | 0.375 | 0.207 | 0.368 |
| | 720 | 0.223 | 0.373 | 0.246 | 0.384 | 0.246 | 0.380 | 0.238 | 0.379 | 0.227 | 0.374 | 0.220 | 0.376 | 0.217 | 0.374 | 0.251 | 0.385 | 0.245 | 0.405 |
| ETTm$_1$ | 24 | 0.016 | 0.091 | 0.014 | 0.087 | 0.015 | 0.090 | 0.015 | 0.089 | 0.013 | 0.085 | 0.017 | 0.092 | 0.015 | 0.091 | 0.015 | 0.092 | 0.020 | 0.107 |
| | 48 | 0.026 | 0.120 | 0.025 | 0.119 | 0.025 | 0.117 | 0.027 | 0.122 | 0.024 | 0.116 | 0.028 | 0.123 | 0.027 | 0.124 | 0.028 | 0.125 | 0.032 | 0.138 |
| | 96 | 0.036 | 0.145 | 0.037 | 0.146 | 0.038 | 0.146 | 0.039 | 0.150 | 0.036 | 0.144 | 0.039 | 0.150 | 0.043 | 0.158 | 0.040 | 0.151 | 0.045 | 0.165 |
| | 288 | 0.063 | 0.191 | 0.074 | 0.205 | 0.072 | 0.204 | 0.072 | 0.205 | 0.080 | 0.216 | 0.078 | 0.211 | 0.082 | 0.218 | 0.075 | 0.205 | 0.093 | 0.238 |
| | 672 | 0.090 | 0.225 | 0.108 | 0.250 | 0.098 | 0.239 | 0.104 | 0.248 | 0.114 | 0.248 | 0.104 | 0.246 | 0.112 | 0.260 | 0.100 | 0.240 | 0.134 | 0.285 |
| Elec. | 24 | 0.240 | 0.266 | 0.244 | 0.266 | 0.241 | 0.267 | 0.242 | 0.264 | 0.243 | 0.272 | 0.241 | 0.264 | 0.240 | 0.264 | 0.242 | 0.265 | 0.251 | 0.279 |
| | 48 | 0.285 | 0.294 | 0.291 | 0.295 | 0.285 | 0.295 | 0.287 | 0.294 | 0.290 | 0.300 | 0.286 | 0.291 | 0.284 | 0.292 | 0.287 | 0.294 | 0.298 | 0.309 |
| | 168 | 0.392 | 0.371 | 0.400 | 0.371 | 0.392 | 0.366 | 0.394 | 0.367 | 0.398 | 0.372 | 0.394 | 0.365 | 0.395 | 0.362 | 0.393 | 0.364 | 0.413 | 0.383 |
| | 336 | 0.542 | 0.461 | 0.547 | 0.465 | 0.541 | 0.464 | 0.542 | 0.461 | 0.541 | 0.470 | 0.545 | 0.460 | 0.545 | 0.457 | 0.539 | 0.457 | 0.541 | 0.460 |
| WTH | 24 | 0.093 | 0.211 | 0.098 | 0.220 | 0.092 | 0.209 | 0.091 | 0.207 | 0.096 | 0.215 | 0.092 | 0.210 | 0.093 | 0.212 | 0.092 | 0.209 | 0.100 | 0.222 |
| | 48 | 0.131 | 0.256 | 0.139 | 0.266 | 0.130 | 0.255 | 0.127 | 0.250 | 0.141 | 0.266 | 0.129 | 0.252 | 0.134 | 0.260 | 0.131 | 0.257 | 0.142 | 0.266 |
| | 168 | 0.182 | 0.311 | 0.194 | 0.324 | 0.185 | 0.314 | 0.184 | 0.316 | 0.208 | 0.336 | 0.186 | 0.315 | 0.195 | 0.327 | 0.185 | 0.315 | 0.207 | 0.335 |
| | 336 | 0.195 | 0.325 | 0.210 | 0.342 | 0.208 | 0.342 | 0.206 | 0.341 | 0.231 | 0.357 | 0.209 | 0.339 | 0.215 | 0.349 | 0.203 | 0.335 | 0.225 | 0.355 |
| | 720 | 0.198 | 0.330 | 0.218 | 0.353 | 0.217 | 0.353 | 0.211 | 0.349 | 0.240 | 0.369 | 0.220 | 0.352 | 0.231 | 0.370 | 0.218 | 0.352 | 0.244 | 0.371 |
| Lora | 24 | 0.052 | 0.141 | 0.060 | 0.152 | 0.138 | 0.219 | 0.128 | 0.213 | 0.078 | 0.171 | 0.067 | 0.170 | 0.158 | 0.223 | 0.057 | 0.154 | 0.070 | 0.185 |
| | 48 | 0.080 | 0.181 | 0.092 | 0.196 | 0.117 | 0.225 | 0.181 | 0.264 | 0.127 | 0.223 | 0.112 | 0.218 | 0.185 | 0.257 | 0.084 | 0.189 | 0.096 | 0.218 |
| | 168 | 0.155 | 0.263 | 0.246 | 0.317 | 0.196 | 0.308 | 0.232 | 0.334 | 0.481 | 0.393 | 0.676 | 0.433 | 0.311 | 0.359 | 0.235 | 0.323 | 0.169 | 0.290 |
| | 336 | 0.229 | 0.335 | 0.302 | 0.372 | 0.395 | 0.444 | 0.363 | 0.433 | 0.941 | 0.532 | 1.403 | 0.619 | 0.378 | 0.429 | 0.535 | 0.475 | 0.245 | 0.353 |
| | 720 | 0.370 | 0.445 | 0.483 | 0.509 | 1.60 | 0.729 | 0.617 | 0.561 | 1.926 | 0.739 | 1.655 | 0.771 | 0.395 | 0.461 | 1.315 | 0.722 | 0.375 | 0.450 |
| Avg. | | 0.157 | 0.258 | 0.173 | 0.270 | 0.216 | 0.282 | 0.185 | 0.280 | 0.260 | 0.294 | 0.266 | 0.394 | 0.184 | 0.281 | 0.212 | 0.284 | 0.176 | 0.286 |

#### D.3.2 VISUALIZATION OF AUGMENTATION

The intuitive understanding of Property 3 is that if the map between x and v is one-to-many, the generated view will contain more information compared to the raw input x. In other words, a good augmentation should preserve the underlying semantics that two different $x$ cannot map the same $v$. In order to further explore the effectiveness of AutoTCL in generating diverse and semantic-preserving views, we used T-SNE to visualize the embeddings of different augmented views in Figure 2. We chose an instance, denoted by $x$, from dataset ETTh$_1$ and compare different augmentation methods, including **Cutout**, **Jitter**, and **Adversarial**. To avoid the special case, we reported 10 augmented views for AutoTCL. We also include another $x'$ instance as a reference. As shown in Figure 2, the instances augmented by AutoTCL include more diversity compared with **Jitter** and **Cutout**. Moreover, the augmentation generated by the **Adversarial** is closer to $x'$ or $x$, indicating that it fails to preserve the underlying semantics.

#### D.3.3 VISUALIZATION OF CONVERGENCE

To show the convergence of our method, we plotted the curves of Eq.( 10) and Eq. (15) on different datasets. As shown in Figure 3, our method converged easily in both the argumentation network and the embedding network. In Figure 3(a) and 3(d), we observed that after the argumentation network converged to a certain level, the encoding network still benefited from that. In Figure 3(b), 3(c), and 3(e), they have the same patterns that the augmentation loss arrived the convergence level almost the

Table 7: Ablation studies using TS2Vec backbone

| Dataset | $L_y$ | AutoTCL MSE | MAE | W/o $h(x)$ MSE | MAE | W/o $g(x)$ MSE | MAE | W/o $\Delta V$ MSE | MAE | W/o Aug MSE | MAE | Cutout MSE | MAE | Jitter MSE | MAE | Adversarial MSE | MAE | Random Aug. MSE | MAE |
|---|---|---|---|---|---|---|---|---|---|---|---|---|---|---|---|---|---|---|---|
| ETTh$_1$ | 24 | **0.039** | **0.146** | **0.039** | 0.148 | 0.047 | 0.165 | 0.047 | 0.164 | **0.039** | 0.149 | 0.043 | 0.155 | 0.040 | 0.150 | 0.041 | 0.151 | 0.047 | 0.166 |
| | 48 | **0.058** | **0.180** | 0.060 | 0.185 | 0.075 | 0.212 | 0.073 | 0.205 | 0.063 | 0.190 | 0.063 | 0.187 | 0.063 | 0.190 | 0.062 | 0.186 | 0.069 | 0.203 |
| | 168 | **0.106** | **0.245** | 0.115 | 0.259 | 0.145 | 0.298 | 0.131 | 0.278 | 0.119 | 0.264 | 0.118 | 0.262 | 0.111 | 0.253 | 0.114 | 0.255 | 0.127 | 0.280 |
| | 336 | **0.121** | **0.266** | 0.139 | 0.289 | 0.159 | 0.317 | 0.148 | 0.303 | 0.141 | 0.291 | 0.133 | 0.285 | 0.130 | 0.277 | 0.132 | 0.280 | 0.140 | 0.300 |
| | 720 | **0.154** | **0.314** | 0.181 | 0.347 | 0.198 | 0.365 | 0.180 | 0.344 | 0.193 | 0.359 | 0.167 | 0.331 | 0.164 | 0.323 | 0.167 | 0.330 | 0.171 | 0.342 |
| ETTh$_2$ | 24 | 0.106 | 0.252 | 0.108 | 0.250 | **0.095** | **0.235** | 0.104 | 0.248 | 0.108 | 0.251 | 0.105 | 0.250 | 0.105 | 0.249 | 0.107 | 0.252 | 0.101 | 0.246 |
| | 48 | 0.131 | 0.284 | 0.134 | 0.285 | **0.129** | **0.279** | 0.136 | 0.289 | 0.140 | 0.290 | 0.133 | 0.287 | 0.135 | 0.287 | 0.137 | 0.288 | 0.129 | 0.281 |
| | 168 | **0.182** | **0.343** | 0.185 | 0.344 | 0.212 | 0.365 | 0.211 | 0.366 | 0.203 | 0.360 | 0.198 | 0.355 | 0.194 | 0.353 | 0.195 | 0.353 | 0.185 | 0.344 |
| | 336 | **0.190** | **0.351** | 0.191 | 0.351 | 0.205 | 0.362 | 0.209 | 0.366 | 0.206 | 0.367 | 0.204 | 0.363 | 0.201 | 0.362 | 0.199 | 0.360 | 0.201 | 0.362 |
| | 720 | 0.204 | 0.370 | 0.204 | 0.368 | 0.203 | 0.366 | 0.200 | **0.364** | 0.205 | 0.369 | 0.205 | 0.367 | 0.200 | **0.364** | 0.194 | 0.359 | 0.234 | 0.329 |
| ETTm$_1$ | 24 | **0.014** | **0.085** | 0.018 | 0.098 | 0.015 | 0.089 | **0.014** | 0.087 | **0.014** | 0.087 | 0.015 | 0.089 | **0.014** | 0.087 | **0.014** | **0.085** | 0.021 | 0.109 |
| | 48 | **0.026** | **0.117** | 0.027 | 0.121 | 0.028 | 0.123 | **0.026** | 0.120 | 0.027 | 0.121 | 0.027 | 0.121 | 0.027 | 0.121 | **0.026** | **0.117** | 0.036 | 0.144 |
| | 96 | **0.038** | **0.147** | 0.039 | 0.149 | 0.039 | 0.147 | 0.041 | 0.153 | 0.041 | 0.153 | 0.041 | 0.153 | **0.038** | **0.147** | 0.038 | 0.147 | 0.050 | 0.171 |
| | 288 | **0.081** | 0.216 | 0.083 | 0.219 | 0.082 | 0.216 | 0.084 | 0.222 | 0.084 | 0.222 | 0.084 | 0.222 | **0.081** | **0.215** | 0.081 | 0.216 | 0.106 | 0.251 |
| | 672 | **0.119** | **0.263** | 0.123 | 0.269 | 0.122 | 0.266 | 0.124 | 0.271 | 0.124 | 0.270 | 0.121 | 0.267 | 0.120 | 0.265 | **0.119** | **0.263** | 0.176 | 0.329 |
| Elec. | 24 | **0.247** | **0.269** | 0.249 | 0.271 | 0.248 | **0.269** | **0.247** | 0.270 | 0.250 | 0.271 | 0.248 | 0.270 | 0.249 | 0.273 | 0.250 | 0.270 | 0.251 | 0.277 |
| | 48 | 0.297 | **0.301** | 0.302 | 0.306 | 0.297 | **0.301** | 0.297 | 0.303 | 0.298 | 0.302 | **0.296** | 0.302 | 0.297 | 0.307 | 0.298 | 0.302 | 0.393 | 0.303 |
| | 168 | **0.408** | 0.380 | 0.413 | 0.381 | **0.408** | 0.380 | 0.410 | 0.380 | 0.408 | 0.377 | 0.408 | 0.383 | 0.410 | 0.384 | **0.408** | **0.377** | 0.405 | 0.372 |
| | 336 | **0.541** | **0.468** | 0.553 | 0.472 | **0.541** | 0.469 | 0.547 | 0.470 | 0.542 | 0.471 | 0.545 | 0.470 | 0.470 | 0.547 | 0.542 | 0.471 | 0.554 | 0.458 |
| WTH | 24 | **0.093** | 0.212 | 0.096 | 0.214 | 0.094 | **0.210** | 0.096 | 0.214 | 0.094 | **0.211** | 0.099 | 0.215 | 0.096 | 0.213 | 0.096 | 0.213 | 0.104 | 0.227 |
| | 48 | 0.133 | 0.258 | 0.134 | 0.259 | **0.130** | **0.253** | 0.134 | 0.258 | 0.134 | 0.257 | 0.132 | 0.256 | 0.135 | 0.259 | 0.133 | 0.254 | 0.142 | 0.268 |
| | 168 | 0.188 | 0.316 | 0.192 | 0.322 | **0.184** | **0.313** | 0.192 | 0.322 | 0.197 | 0.324 | 0.189 | 0.317 | 0.192 | 0.321 | 0.193 | 0.322 | 0.195 | 0.323 |
| | 336 | **0.201** | **0.333** | 0.208 | 0.341 | 0.202 | 0.335 | 0.208 | 0.341 | 0.216 | 0.347 | 0.208 | 0.338 | 0.211 | 0.342 | 0.212 | 0.344 | 0.209 | 0.340 |
| | 720 | **0.204** | **0.339** | 0.210 | 0.347 | 0.204 | 0.339 | 0.210 | 0.347 | 0.225 | 0.357 | 0.211 | 0.343 | 0.220 | 0.353 | 0.217 | 0.352 | 0.211 | 0.344 |
| Lora | 24 | **0.053** | **0.140** | 0.061 | 0.156 | 0.076 | 0.169 | 0.061 | 0.156 | 0.188 | 0.219 | 0.062 | 0.158 | 0.080 | 0.173 | 0.062 | 0.157 | 0.076 | 0.177 |
| | 48 | **0.082** | **0.188** | 0.087 | 0.193 | 0.136 | 0.229 | 0.086 | 0.193 | 0.296 | 0.285 | 0.086 | 0.192 | 0.155 | 0.238 | 0.088 | 0.195 | 0.115 | 0.222 |
| | 168 | **0.161** | **0.278** | 0.210 | 0.306 | 0.210 | 0.308 | 0.344 | 0.360 | 0.341 | 0.371 | 0.188 | 0.294 | 0.229 | 0.318 | 0.219 | 0.310 | 0.219 | 0.321 |
| | 336 | **0.231** | **0.347** | 0.454 | 0.431 | 0.274 | 0.369 | 0.299 | 0.391 | 0.400 | 0.420 | 0.369 | 0.407 | 0.281 | 0.373 | 0.488 | 0.438 | 0.323 | 0.401 |
| | 720 | 0.375 | 0.451 | 0.369 | 0.447 | 0.400 | 0.473 | 0.370 | 0.428 | 0.484 | 0.482 | 0.367 | **0.446** | 0.443 | 0.493 | **0.364** | 0.454 | 0.465 | 0.512 |
| Avg. | | **0.165** | **0.271** | 0.179 | 0.280 | 0.178 | 0.284 | 0.180 | 0.283 | 0.199 | 0.291 | 0.175 | 0.279 | 0.176 | 0.284 | 0.179 | 0.279 | 0.185 | 0.292 |

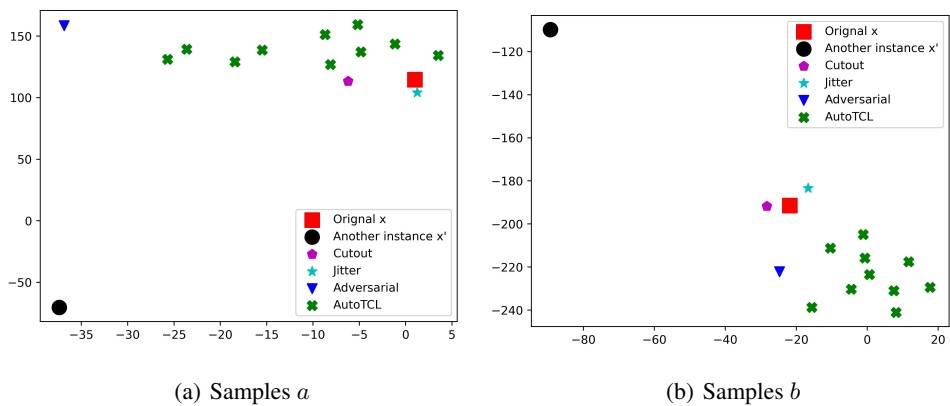

(a) Samples $a$          (b) Samples $b$

Figure 2: T-SNE visualization of different augmentation instances. In samples $a$ and $b$, AutoTCL-generated samples are closer to the original instance $x$ than other instances $x'$ with large variety

same as the contrastive loss. While the situation was different in Figure 3(f), at the beginning the augmentation network benefited from encoding loss, then two losses converged gradually.

### D.3.4 PARAMETER SENSITIVITY STUDIES

In the proposed AutoTCL, we have three hyper-parameters, $\alpha$ in Eq. (15), $\beta$ in Eq. (8), and $\gamma$ in Eq. (10), to get the trade-off in training the augmentation network and the encoder network. In this part, we chose different values for these three variables in the range from 0.0001 to 0.3 and reported MSE and MAE scores in the ETTh1 dataset. The results of this part could be found in Figure 4. The sensitivity studies result of three hyper-parameters are shown in Figure 4. From this figure, some results could be observed that our method is able to achieve comparative performances with a wide range of choices for these three hyperparameters, indicating the robustness of our method. The $\beta$ in

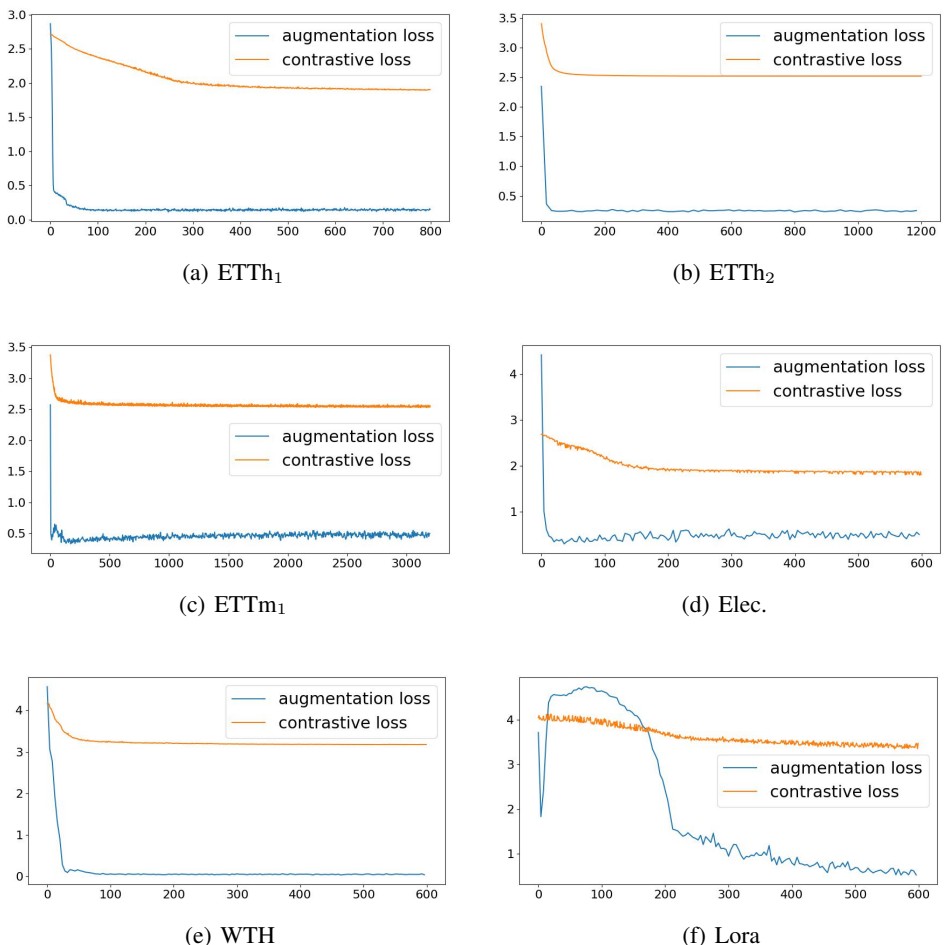

Figure 3: The augmentation loss, Eq. (10) and contrastive loss, Eq. (15), in the training process

Eq. (8), and $\gamma$ in Eq. (10) have the opposite effect as the weight goes up. Second, we observe that small values, such as 0.001, give good performances on ETTh1 datasets as well as others.

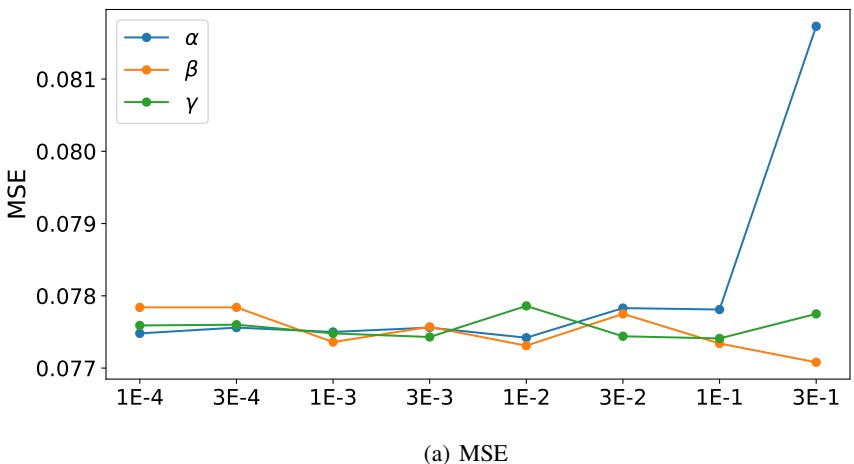

(a) MSE

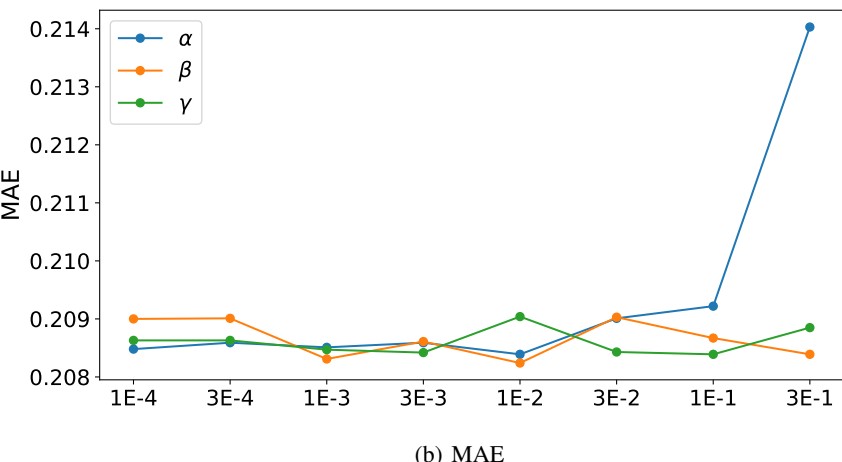

(b) MAE

Figure 4: Parameter sensitivity studies on ETTh$_1$.

### D.3.5  CASE STUDY

To further explore the augmentation of AutoTCL, we have done the case study in this section. We selected four instances to show the effectiveness of our method in Table 9. As shown in Table 9, we used the CricketX dataset as input instances and got the informative part by using the augmentation network to get masks, the result of $h(x)$. With the regularization loss help in Eq. (9), our method could have a continuous mask that makes the informative part more consistent. From the results, the informative parts detected by $h()$ appear to retain the prominent peaks and troughs of the original sequence, which are typically considered significant by domain experts. In time series analysis, such prominent features often correspond to critical events or changes in the underlying system's behavior.

### D.3.6  EXPERIMENTS ON OTHER DOMAIN DATASET

In order to further verify the adaptability of our method, we conducted experiments on Traffic[4] dataset. This dataset comprises hourly information sourced from the California Department of Transportation.

---

[4]https://pems.dot.ca.gov/

This dataset delineates road occupancy rates as observed by various sensors deployed on the freeways in the San Francisco Bay area. Following the default setting, we adopt CoST as the backbone and conduct forecasting in both univariate and multivariate forecasting settings. We also include another SOTA method, TS2Vec as a comparison. The results are shown in Table 8. We observe that, in both univariate and multivariate forecasting settings, AutoTCL achieves the best results in all prediction lengths. On average, AutoTCL decreases MSE by $9.4\%$, MAE by $5.7\%$ in the univariate forecasting setting and MSE by $5.0\%$, MAE by $9.3\%$ in the multivariate forecasting setting comparing to the baseline.

Table 8: Forecasting results on Traffic dataset.

| Settings | $L_y$ | AutoTCL | | TS2Vec | | CoST | |
|---|---|---|---|---|---|---|---|
| | | MSE | MAE | MSE | MAE | MSE | MAE |
| Univariate | 96 | **0.253** | **0.353** | 0.431 | 0.484 | 0.284 | 0.379 |
| | 192 | **0.271** | **0.373** | 0.437 | 0.489 | 0.302 | 0.398 |
| | 336 | **0.312** | **0.414** | 0.453 | 0.500 | 0.340 | 0.435 |
| | 720 | **0.357** | **0.447** | 0.464 | 0.508 | 0.390 | 0.474 |
| | Avg. | **0.298** | **0.397** | 0.446 | 0.495 | 0.329 | 0.421 |
| Multivariate | 96 | **0.715** | **0.396** | 1.038 | 0.574 | 0.759 | 0.442 |
| | 192 | **0.722** | **0.396** | 1.042 | 0.588 | 0.757 | 0.434 |
| | 336 | **0.730** | **0.396** | 1.064 | 0.594 | 0.765 | 0.435 |
| | 720 | **0.746** | **0.403** | 1.085 | 0.604 | 0.784 | 0.444 |
| | Avg. | **0.728** | **0.398** | 1.057 | 0.590 | 0.766 | 0.439 |

## D.4 FULL EXPERIMENTS

**Univariate forecasting.** Full experiment results of univariate time series forecasting results can be found in Table 10. In these experiments, AutoTCL achieved minimum error in most cases. Compared to the state-of-the-art CoST method, AutoTCL reduced the average MSE error by $6.5\%$ and the average MAE by $4.8\%$.

**Multivariate forecasting.** We provided our full experiment results of multivariate time series forecasting results in Table 11. In multivariate forecasting tasks, our method achieved fewer best results than univariate forecasting. AutoTCL reduced the average MSE error by $2.9\%$ and the average MAE by $1.2\%$ than CoST. In the column of stemGNN, because of the error out-of-memory, we can't report part results.

**Classification.** In Table 12, the full results of 30 class datasets are provided. AutoTCL is the most powerful method than other baselines with the highest average accuracy rate and ranks. Some results are not available due to out-of-memory errors.

Table 9: Case study of parametric augmentation. The inputs are from the CricketX dataset, which is a univariate time series dataset. We demonstrate the informative parts in two settings, w/ and w/o ($\gamma = 0$) regularization loss.

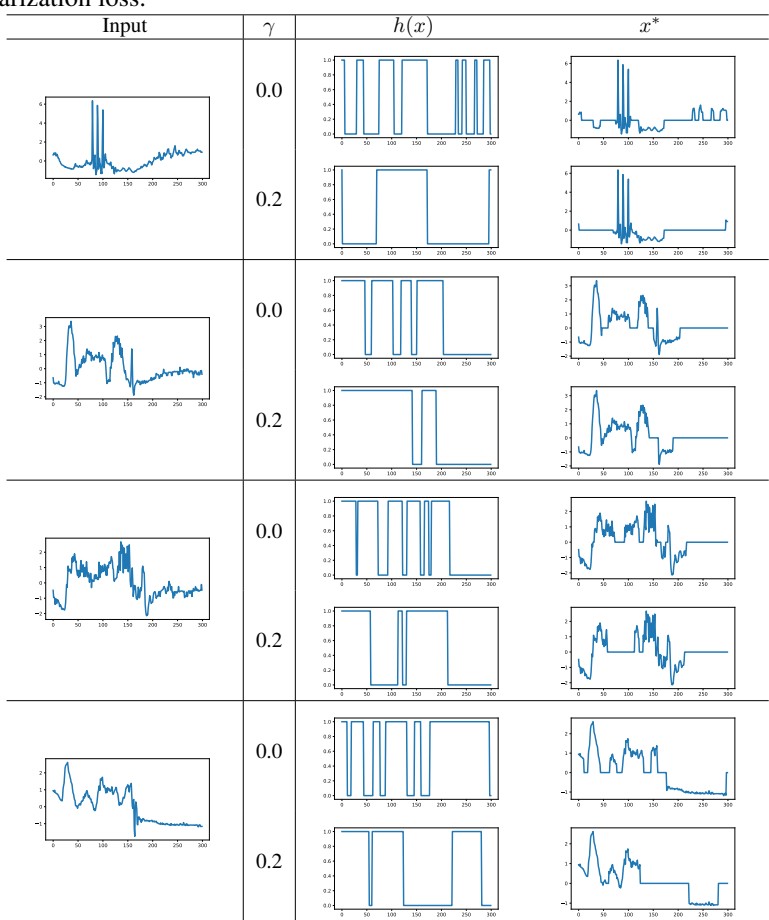

Table 10: Univariate time series forecasting results.

| Dataset | $L_y$ | AutoTCL | | TS2Vec | | Informer | | LogTrans | | N-BEATS | | TCN | | CoST | | TNC | | TS-TCC | | InfoTS | |
|---|---|---|---|---|---|---|---|---|---|---|---|---|---|---|---|---|---|---|---|---|---|
| | | MSE | MAE | MSE | MAE | MSE | MAE | MSE | MAE | MSE | MAE | MSE | MAE | MSE | MAE | MSE | MAE | MSE | MAE | MSE | MAE |
| ETTh1 | 24 | **0.037** | **0.148** | 0.039 | 0.152 | 0.098 | 0.247 | 0.103 | 0.259 | 0.094 | 0.238 | 0.075 | 0.210 | 0.040 | 0.152 | 0.057 | 0.184 | 0.103 | 0.237 | 0.039 | 0.149 |
| | 48 | **0.054** | **0.176** | 0.062 | 0.191 | 0.158 | 0.319 | 0.167 | 0.328 | 0.210 | 0.367 | 0.227 | 0.402 | 0.060 | 0.186 | 0.094 | 0.239 | 0.139 | 0.279 | 0.056 | 0.179 |
| | 168 | **0.078** | **0.210** | 0.134 | 0.282 | 0.183 | 0.346 | 0.207 | 0.375 | 0.232 | 0.391 | 0.316 | 0.493 | 0.097 | 0.236 | 0.171 | 0.329 | 0.253 | 0.408 | 0.100 | 0.239 |
| | 336 | **0.093** | **0.231** | 0.154 | 0.310 | 0.222 | 0.387 | 0.230 | 0.398 | 0.232 | 0.388 | 0.306 | 0.495 | 0.112 | 0.258 | 0.192 | 0.357 | 0.155 | 0.318 | 0.117 | 0.264 |
| | 720 | **0.120** | **0.272** | 0.163 | 0.327 | 0.269 | 0.435 | 0.273 | 0.463 | 0.322 | 0.490 | 0.390 | 0.557 | 0.148 | 0.306 | 0.235 | 0.408 | 0.190 | 0.337 | 0.141 | 0.302 |
| ETTh2 | 24 | **0.079** | **0.206** | 0.090 | 0.229 | 0.093 | 0.240 | 0.102 | 0.255 | 0.198 | 0.345 | 0.103 | 0.249 | 0.079 | 0.207 | 0.097 | 0.238 | 0.239 | 0.391 | 0.081 | 0.215 |
| | 48 | 0.117 | 0.255 | 0.124 | 0.273 | 0.155 | 0.314 | 0.169 | 0.348 | 0.234 | 0.386 | 0.142 | 0.290 | 0.118 | 0.259 | 0.131 | 0.281 | 0.260 | 0.405 | 0.115 | 0.261 |
| | 168 | 0.176 | **0.319** | 0.208 | 0.360 | 0.232 | 0.389 | 0.246 | 0.422 | 0.331 | 0.453 | 0.227 | 0.376 | 0.189 | 0.339 | 0.197 | 0.354 | 0.291 | 0.420 | **0.171** | 0.327 |
| | 336 | 0.193 | 0.344 | 0.213 | 0.369 | 0.263 | 0.417 | 0.267 | 0.437 | 0.431 | 0.508 | 0.296 | 0.430 | 0.206 | 0.360 | 0.207 | 0.366 | 0.336 | 0.453 | **0.183** | **0.341** |
| | 720 | 0.223 | 0.373 | 0.214 | 0.374 | 0.277 | 0.431 | 0.303 | 0.493 | 0.437 | 0.517 | 0.325 | 0.463 | 0.214 | 0.371 | 0.207 | 0.370 | 0.362 | 0.472 | **0.194** | **0.357** |
| ETTm1 | 24 | 0.016 | 0.091 | 0.015 | 0.092 | 0.030 | 0.137 | 0.065 | 0.202 | 0.054 | 0.184 | 0.041 | 0.157 | 0.015 | 0.088 | 0.019 | 0.103 | 0.089 | 0.228 | **0.014** | **0.087** |
| | 48 | 0.026 | 0.120 | 0.027 | 0.126 | 0.069 | 0.203 | 0.078 | 0.220 | 0.190 | 0.361 | 0.101 | 0.257 | **0.025** | **0.117** | 0.036 | 0.142 | 0.134 | 0.280 | **0.025** | **0.117** |
| | 96 | **0.036** | **0.145** | 0.044 | 0.161 | 0.194 | 0.372 | 0.199 | 0.386 | 0.183 | 0.353 | 0.142 | 0.311 | 0.038 | 0.147 | 0.054 | 0.178 | 0.159 | 0.305 | **0.036** | 0.142 |
| | 288 | **0.063** | **0.191** | 0.103 | 0.246 | 0.401 | 0.554 | 0.411 | 0.572 | 0.186 | 0.362 | 0.318 | 0.472 | 0.077 | 0.209 | 0.098 | 0.244 | 0.204 | 0.327 | 0.071 | 0.200 |
| | 672 | **0.090** | **0.225** | 0.156 | 0.307 | 0.512 | 0.644 | 0.598 | 0.702 | 0.197 | 0.368 | 0.397 | 0.547 | 0.113 | 0.257 | 0.136 | 0.290 | 0.206 | 0.354 | 0.102 | 0.240 |
| Elec. | 24 | **0.241** | **0.262** | 0.260 | 0.288 | 0.251 | 0.275 | 0.528 | 0.447 | 0.427 | 0.330 | 0.263 | 0.279 | 0.243 | 0.264 | 0.252 | 0.278 | 0.379 | 0.561 | 0.245 | 0.269 |
| | 48 | **0.287** | **0.292** | 0.319 | 0.324 | 0.346 | 0.339 | 0.409 | 0.414 | 0.551 | 0.392 | 0.373 | 0.344 | 0.292 | 0.300 | 0.300 | 0.308 | 0.453 | 0.600 | 0.294 | 0.301 |
| | 168 | **0.394** | **0.365** | 0.427 | 0.394 | 0.544 | 0.424 | 0.959 | 0.612 | 0.893 | 0.538 | 0.609 | 0.462 | 0.405 | 0.375 | 0.412 | 0.384 | 0.575 | 0.616 | 0.402 | 0.367 |
| | 336 | 0.543 | 0.460 | 0.565 | 0.474 | 0.713 | 0.512 | 1.079 | 0.639 | 1.035 | 0.669 | 0.855 | 0.606 | 0.560 | 0.473 | 0.548 | 0.466 | 0.637 | 0.633 | **0.533** | **0.453** |
| WTH | 24 | **0.093** | **0.211** | 0.096 | 0.215 | 0.117 | 0.251 | 0.136 | 0.279 | 0.136 | 0.264 | 0.109 | 0.217 | 0.096 | 0.213 | 0.102 | 0.221 | 0.221 | 0.386 | 0.101 | 0.222 |
| | 48 | **0.131** | **0.256** | 0.140 | 0.264 | 0.178 | 0.318 | 0.206 | 0.356 | 0.198 | 0.319 | 0.143 | 0.269 | 0.138 | 0.262 | 0.139 | 0.264 | 0.255 | 0.406 | 0.141 | 0.266 |
| | 168 | **0.182** | **0.311** | 0.207 | 0.335 | 0.266 | 0.398 | 0.309 | 0.439 | 0.309 | 0.420 | 0.188 | 0.319 | 0.207 | 0.334 | 0.198 | 0.328 | 0.339 | 0.458 | 0.199 | 0.328 |
| | 336 | 0.195 | 0.325 | 0.231 | 0.360 | 0.297 | 0.416 | 0.359 | 0.484 | 0.369 | 0.460 | **0.192** | **0.320** | 0.230 | 0.356 | 0.215 | 0.347 | 0.372 | 0.491 | 0.220 | 0.351 |
| | 720 | **0.198** | 0.330 | 0.233 | 0.365 | 0.359 | 0.466 | 0.388 | 0.499 | 0.270 | 0.406 | **0.198** | **0.329** | 0.242 | 0.370 | 0.219 | 0.353 | 0.322 | 0.467 | 0.218 | 0.353 |
| Lora | 24 | **0.052** | **0.141** | 0.212 | 0.268 | 0.917 | 0.720 | 0.264 | 0.371 | 0.072 | 0.170 | 0.981 | 0.899 | 0.053 | 0.144 | 0.206 | 0.273 | 0.365 | 0.514 | 0.058 | 0.149 |
| | 48 | **0.080** | **0.181** | 0.267 | 0.316 | 1.067 | 0.786 | 0.364 | 0.424 | 0.115 | 0.223 | 0.981 | 0.898 | 0.082 | 0.184 | 0.286 | 0.349 | 0.426 | 0.562 | 0.090 | 0.192 |
| | 168 | **0.155** | **0.263** | 0.355 | 0.389 | 1.745 | 1.067 | 0.452 | 0.465 | 0.286 | 0.350 | 1.276 | 0.946 | 0.166 | 0.274 | 0.523 | 0.549 | 0.481 | 0.587 | 0.156 | 0.267 |
| | 336 | **0.229** | **0.335** | 0.425 | 0.441 | 1.661 | 1.050 | 0.950 | 0.683 | 0.405 | 0.429 | 1.273 | 0.943 | 0.252 | 0.355 | 0.772 | 0.724 | 0.588 | 0.645 | 0.313 | 0.386 |
| | 720 | **0.370** | **0.445** | 0.523 | 0.509 | 2.482 | 1.370 | 1.248 | 0.807 | 0.679 | 0.573 | 1.290 | 0.950 | 0.379 | 0.451 | 1.313 | 0.929 | 0.592 | 0.649 | 1.047 | 0.635 |
| Avg. | | **0.157** | **0.258** | 0.207 | 0.301 | 0.486 | 0.477 | 0.382 | 0.441 | 0.320 | 0.388 | 0.419 | 0.465 | 0.168 | 0.271 | 0.256 | 0.340 | 0.315 | 0.441 | 0.188 | 0.274 |

Table 11: Multivariate time series forecasting results.

| Dataset | $L_y$ | AutoTCL | | TS2Vec | | Informer | | LogTrans | | StemGNN | | TCN | | CoST | | TNC | | TS-TCC | | InfoTS | |
|---|---|---|---|---|---|---|---|---|---|---|---|---|---|---|---|---|---|---|---|---|---|
| | | MSE | MAE | MSE | MAE | MSE | MAE | MSE | MAE | MSE | MAE | MSE | MAE | MSE | MAE | MSE | MAE | MSE | MAE | MSE | MAE |
| ETTh1 | 24 | 0.389 | 0.439 | 0.599 | 0.534 | 0.577 | 0.549 | 0.686 | 0.604 | 0.614 | 0.571 | 0.767 | 0.612 | **0.386** | **0.429** | 0.708 | 0.592 | 0.516 | 0.508 | 0.564 | 0.520 |
| | 48 | 0.447 | 0.477 | 0.629 | 0.555 | 0.685 | 0.625 | 0.766 | 0.757 | 0.748 | 0.618 | 0.713 | 0.617 | **0.437** | **0.464** | 0.749 | 0.619 | 0.644 | 0.579 | 0.607 | 0.553 |
| | 168 | **0.615** | **0.574** | 0.755 | 0.636 | 0.931 | 0.752 | 1.002 | 0.846 | 0.663 | 0.608 | 0.995 | 0.738 | 0.643 | 0.582 | 0.884 | 0.699 | 0.678 | 0.619 | 0.746 | 0.638 |
| | 336 | **0.802** | **0.671** | 0.907 | 0.717 | 1.128 | 0.873 | 1.362 | 0.952 | 0.800 | 0.812 | 1.175 | 0.800 | 0.812 | 0.679 | 1.020 | 0.768 | 0.967 | 0.754 | 0.904 | 0.722 |
| | 720 | 1.028 | 0.789 | 1.048 | 0.790 | 1.215 | 0.896 | 1.397 | 1.291 | – | – | 1.453 | 1.311 | 0.970 | 0.771 | 1.157 | 0.830 | **0.935** | **0.715** | 1.098 | 0.811 |
| ETTh2 | 24 | **0.337** | **0.433** | 0.398 | 0.461 | 0.720 | 0.665 | 0.828 | 0.750 | 1.292 | 0.883 | 1.365 | 0.888 | 0.447 | 0.502 | 0.612 | 0.595 | 0.782 | 0.666 | 0.383 | 0.462 |
| | 48 | 0.572 | 0.576 | 0.578 | 0.573 | 1.457 | 1.001 | 1.806 | 1.034 | 1.099 | 0.847 | 1.395 | 0.960 | 0.637 | 0.637 | 0.840 | 0.716 | 1.357 | 0.881 | **0.567** | **0.582** |
| | 168 | **1.470** | **0.947** | 1.901 | 1.065 | 3.489 | 1.515 | 4.070 | 1.681 | 2.282 | 1.228 | 3.166 | 1.407 | 1.549 | 0.982 | 2.359 | 1.213 | 4.318 | 1.728 | 1.789 | 1.048 |
| | 336 | **1.685** | **1.027** | 2.304 | 1.215 | 2.723 | 1.340 | 3.875 | 1.763 | 3.086 | 1.351 | 3.256 | 1.481 | 1.749 | 1.042 | 2.782 | 1.349 | 2.097 | 1.145 | 2.120 | 1.161 |
| | 720 | **1.890** | **1.092** | 2.650 | 1.373 | 2.723 | 1.473 | 3.913 | 1.552 | – | – | 3.690 | 1.588 | 1.971 | 1.092 | 2.753 | 1.394 | 2.047 | 1.127 | 2.511 | 1.316 |
| ETTm1 | 24 | 0.256 | 0.339 | 0.443 | 0.436 | 0.323 | 0.369 | 0.419 | 0.412 | 0.620 | 0.570 | 0.324 | 0.374 | **0.246** | **0.329** | 0.522 | 0.472 | 0.403 | 0.455 | 0.391 | 0.408 |
| | 48 | 0.339 | 0.396 | 0.582 | 0.515 | 0.494 | 0.503 | 0.507 | 0.583 | 0.744 | 0.628 | 0.477 | 0.450 | **0.331** | **0.386** | 0.695 | 0.567 | 0.618 | 0.552 | 0.503 | 0.475 |
| | 96 | 0.376 | 0.422 | 0.622 | 0.549 | 0.678 | 0.614 | 0.768 | 0.792 | 0.620 | 0.624 | 0.636 | 0.602 | **0.378** | **0.419** | 0.731 | 0.595 | 0.607 | 0.572 | 0.537 | 0.503 |
| | 288 | **0.464** | **0.484** | 0.709 | 0.609 | 1.056 | 0.786 | 1.462 | 1.320 | 0.843 | 0.683 | 1.270 | 1.351 | 0.472 | 0.486 | 0.818 | 0.649 | 0.722 | 0.638 | 0.653 | 0.579 |
| | 672 | **0.608** | **0.566** | 0.786 | 0.655 | 1.192 | 0.926 | 1.669 | 1.461 | – | – | 1.381 | 1.467 | 0.620 | 0.574 | 0.932 | 0.712 | 0.708 | 0.601 | 0.757 | 0.642 |
| Elec. | 24 | 0.153 | 0.250 | 0.287 | 0.374 | 0.312 | 0.387 | 0.297 | 0.374 | 0.439 | 0.388 | 0.305 | 0.384 | **0.136** | **0.242** | 0.354 | 0.423 | 0.379 | 0.561 | 0.255 | 0.350 |
| | 48 | 0.167 | 0.264 | 0.307 | 0.388 | 0.392 | 0.431 | 0.316 | 0.389 | 0.413 | 0.455 | 0.317 | 0.392 | **0.153** | **0.258** | 0.376 | 0.438 | 0.453 | 0.600 | 0.279 | 0.368 |
| | 168 | 0.179 | 0.275 | 0.332 | 0.407 | 0.515 | 0.509 | 0.426 | 0.466 | 0.506 | 0.518 | 0.358 | 0.423 | **0.175** | **0.275** | 0.402 | 0.456 | 0.575 | 0.616 | 0.302 | 0.385 |
| | 336 | 0.199 | 0.297 | 0.349 | 0.420 | 0.759 | 0.625 | 0.365 | 0.417 | 0.647 | 0.596 | 0.349 | 0.416 | **0.196** | **0.296** | 0.417 | 0.466 | 0.637 | 0.633 | 0.320 | 0.399 |
| WTH | 24 | 0.302 | 0.364 | 0.307 | 0.363 | 0.335 | 0.381 | 0.435 | 0.477 | **0.283** | 0.507 | 0.321 | 0.367 | 0.298 | 0.360 | 0.320 | 0.373 | 0.356 | 0.463 | 0.316 | 0.369 |
| | 48 | 0.361 | 0.412 | 0.374 | 0.418 | 0.395 | 0.459 | 0.426 | 0.495 | **0.337** | 0.573 | 0.386 | 0.423 | 0.359 | **0.411** | 0.380 | 0.421 | 0.429 | 0.500 | 0.381 | 0.420 |
| | 168 | 0.455 | 0.484 | 0.491 | 0.506 | 0.608 | 0.567 | 0.727 | 0.671 | **0.397** | 0.652 | 0.491 | 0.501 | 0.464 | 0.491 | 0.479 | 0.495 | 0.511 | 0.550 | 0.490 | 0.501 |
| | 336 | 0.487 | 0.505 | 0.525 | 0.530 | 0.702 | 0.620 | 0.754 | 0.670 | **0.394** | 0.639 | 0.502 | 0.507 | 0.497 | 0.517 | 0.505 | 0.514 | 0.575 | 0.584 | 0.532 | 0.527 |
| | 720 | 0.508 | 0.519 | 0.556 | 0.552 | 0.831 | 0.731 | 0.885 | 0.773 | – | – | **0.498** | **0.508** | 0.533 | 0.542 | 0.519 | 0.525 | 0.545 | 0.577 | 0.554 | 0.543 |
| Lora | 24 | 0.198 | 0.252 | 0.212 | 0.267 | 0.376 | 0.345 | 0.456 | 0.394 | **0.161** | 0.373 | 0.854 | 0.775 | 0.202 | 0.259 | 0.264 | 0.302 | 0.365 | 0.514 | 0.198 | **0.243** |
| | 48 | 0.254 | 0.301 | 0.266 | 0.316 | 0.428 | 0.420 | 0.663 | 0.467 | **0.204** | 0.439 | 0.851 | 0.774 | 0.258 | 0.307 | 0.319 | 0.345 | 0.426 | 0.562 | 0.254 | **0.297** |
| | 168 | 0.346 | 0.377 | 0.354 | 0.389 | 0.734 | 0.597 | 0.682 | 0.510 | **0.270** | 0.536 | 1.118 | 0.839 | 0.350 | 0.383 | 0.474 | 0.477 | 0.481 | 0.587 | 0.345 | 0.374 |
| | 336 | 0.414 | 0.428 | 0.425 | 0.441 | 0.995 | 0.738 | 1.068 | 0.608 | **0.395** | 0.618 | 1.111 | 0.836 | 0.417 | 0.432 | 0.625 | 0.588 | 0.588 | 0.645 | 0.412 | 0.427 |
| | 720 | 0.517 | 0.502 | 0.522 | 0.509 | 1.181 | 0.831 | 0.959 | 0.622 | – | – | 1.131 | 0.844 | 0.524 | 0.507 | 1.266 | 0.876 | 0.592 | 0.649 | **0.514** | **0.501** |
| Avg. | | **0.545** | **0.499** | 0.697 | 0.571 | 0.990 | 0.708 | 1.138 | 0.798 | 0.753 | 0.651 | 1.057 | 0.781 | 0.561 | 0.505 | 0.837 | 0.637 | 0.838 | 0.675 | 0.665 | 0.556 |

Table 12: Classification result of the UEA dataset

| Dataset | AutoTCL | TS2Vec | T-Loss | TNC | TS-TCC | TST | DTW | TF-C | InfoTS |
|---------|---------|--------|--------|-----|--------|-----|-----|------|--------|
| ArticularyWordRecognition | 0.983 | 0.987 | 0.943 | 0.973 | 0.953 | 0.977 | 0.987 | 0.467 | **0.993** |
| AtrialFibrillation | **0.467** | 0.200 | 0.133 | 0.133 | 0.267 | 0.067 | 0.200 | 0.040 | 0.267 |
| BasicMotions | **1.000** | 0.975 | **1.000** | 0.975 | **1.000** | 0.975 | 0.975 | 0.475 | **1.000** |
| CharacterTrajectories | 0.976 | **0.995** | 0.993 | 0.967 | 0.985 | 0.975 | 0.989 | 0.090 | 0.987 |
| Cricket | **1.000** | 0.972 | 0.972 | 0.958 | 0.917 | **1.000** | **1.000** | 0.125 | **1.000** |
| DuckDuckGeese | **0.700** | 0.680 | 0.650 | 0.460 | 0.380 | 0.620 | 0.600 | 0.340 | 0.600 |
| EigenWorms | **0.901** | 0.847 | 0.840 | 0.840 | 0.779 | 0.748 | 0.618 | – | 0.748 |
| Epilepsy | 0.978 | 0.964 | 0.971 | 0.957 | 0.957 | 0.949 | 0.964 | 0.217 | **0.993** |
| ERing | 0.944 | 0.874 | 0.133 | 0.852 | 0.904 | 0.874 | 0.133 | 0.167 | **0.953** |
| EthanolConcentration | **0.354** | 0.308 | 0.205 | 0.297 | 0.285 | 0.262 | 0.323 | 0.247 | 0.323 |
| FaceDetection | **0.581** | 0.501 | 0.513 | 0.536 | 0.544 | 0.534 | 0.529 | 0.502 | 0.525 |
| FingerMovements | **0.640** | 0.480 | 0.580 | 0.470 | 0.460 | 0.560 | 0.530 | 0.510 | 0.620 |
| HandMovementDirection | 0.432 | 0.338 | 0.351 | 0.324 | 0.243 | 0.243 | 0.231 | 0.405 | **0.514** |
| Handwriting | 0.384 | 0.515 | 0.451 | 0.249 | 0.498 | 0.225 | 0.286 | 0.051 | **0.554** |
| Heartbeat | **0.785** | 0.683 | 0.741 | 0.746 | 0.751 | 0.746 | 0.717 | 0.737 | 0.771 |
| JapaneseVowels | 0.984 | 0.984 | **0.989** | 0.978 | 0.930 | 0.978 | 0.949 | 0.135 | 0.986 |
| Libras | 0.833 | 0.867 | 0.883 | 0.822 | 0.817 | 0.656 | 0.870 | 0.067 | **0.889** |
| LSST | 0.554 | 0.537 | 0.509 | **0.595** | 0.474 | 0.408 | 0.551 | 0.314 | 0.593 |
| MotorImagery | 0.570 | 0.510 | 0.580 | 0.500 | **0.610** | 0.500 | 0.500 | 0.500 | **0.610** |
| NATOPS | **0.944** | 0.928 | 0.917 | 0.911 | 0.822 | 0.850 | 0.883 | 0.533 | 0.939 |
| PEMS-SF | **0.838** | 0.682 | 0.676 | 0.699 | 0.734 | 0.740 | 0.711 | 0.312 | 0.757 |
| PenDigits | 0.984 | **0.989** | 0.981 | 0.979 | 0.974 | 0.560 | 0.977 | 0.236 | **0.989** |
| PhonemeSpectra | 0.218 | 0.233 | 0.222 | 0.207 | **0.252** | 0.085 | 0.151 | 0.026 | 0.233 |
| RacketSports | **0.914** | 0.855 | 0.855 | 0.776 | 0.816 | 0.809 | 0.803 | 0.480 | 0.829 |
| SelfRegulationSCP1 | **0.891** | 0.812 | 0.843 | 0.799 | 0.823 | 0.754 | 0.775 | 0.502 | 0.887 |
| SelfRegulationSCP2 | **0.578** | **0.578** | 0.539 | 0.550 | 0.533 | 0.550 | 0.539 | 0.500 | 0.527 |
| SpokenArabicDigits | 0.925 | 0.932 | 0.905 | 0.934 | 0.970 | 0.923 | 0.963 | 0.100 | **0.988** |
| StandWalkJump | **0.533** | 0.467 | 0.333 | 0.400 | 0.333 | 0.267 | 0.200 | 0.333 | 0.467 |
| UWaveGestureLibrary | 0.893 | 0.884 | 0.875 | 0.759 | 0.753 | 0.575 | 0.903 | 0.125 | **0.906** |
| InsectWingbeat | **0.488** | 0.466 | 0.156 | 0.469 | 0.264 | 0.105 | – | 0.108 | 0.472 |
| Avg. ACC | **0.742** | 0.704 | 0.658 | 0.670 | 0.668 | 0.617 | 0.629 | 0.298 | 0.730 |
| Avg. RANK | **2.300** | 3.700 | 4.667 | 5.433 | 5.133 | 6.133 | 5.400 | 8.200 | 2.367 |

