# OpenReview forum: "Parametric Augmentation for Time Series Contrastive Learning"
_ICLR.cc/2024/Conference — ICLR 2024 poster_

### Official Review · Reviewer_NeuG · 2023-10-27

**Soundness:** 3 good
**Presentation:** 3 good
**Contribution:** 2 fair
**Rating:** 6
**Confidence:** 3

**Summary:**

The study addresses the challenge of data augmentation in time series contrastive learning, where traditional human-guided augmentations might not be directly applicable. The authors analyze time series data augmentation using an information theory perspective and introduce a new parametric augmentation method, named AutoTCL. This method adaptively creates augmentations for time series contrastive learning without relying on pre-defined knowledge. The proposed approach is encoder-independent, making it compatible with various backbone encoders. Experimental results indicate that AutoTCL outperforms leading baselines in both univariate and multivariate forecasting tasks, as well as in classification tasks.

**Strengths:**

S1. A Well-Written Paper: the motivation, method, and experiments are illustrated clearly,

S2. Novel Approach: The introduction of AutoTCL, a parametric augmentation method, offers an adaptive solution to time series contrastive learning, filling a significant gap in existing research.

S3. Novel and Automatic Augmentation Learning: Unlike traditional methods that rely on human intuition or domain knowledge, AutoTCL can automatically learn effective augmentations for time series data, reducing the need for manual tuning or trial-and-error approaches.

S4. Theoretical Foundation: The paper analyzes time series data augmentation from an information theory perspective, providing a good theoretical underpinning for its methodology.

**Weaknesses:**

W1. Support of Experiments:
My main concern is that the authors of this paper claim AutoTCL is a general augmentation method. While the method has been empirically proven effective in crossing different datasets, it would be very critical to test the methods on other contrastive learning frameworks.


W2. Discussion of Related Work:
The authors have given sufficient discussion about augmentation techniques in the original data space. Recently, another branch of augmentations has stemmed from the feature space. I'd also suggested authors to discussion them for a more comprehensive review. [1,2,3]

[1] Towards domain-agnostic contrastive learning

[2] Metaug: Contrastive learning via meta feature augmentation

[3] Hallucination Improves the Performance of Unsupervised Visual Representation Learning

**Questions:**

The main questions are listed in Weaknesses. I'd raise my score if they were appropriately addressed.

The performance of Multivariate time series forecasting results seems to be sub-optimal compared with univariate time. Could the authors provide some insights?

---

> ### Author Response · Authors · 2023-11-21
> **Official response to Reviewer NeuG**
>
> Dear reviewer NeuG,
>
> thank you for taking the time to review our work and providing feedback. In the following, we aim to address your questions and concerns.
>
> > W1. Support of Experiments: My main concern is that the authors of this paper claim AutoTCL is a general augmentation method. While the method has been empirically proven effective in crossing different datasets, it would be very critical to test the methods on other contrastive learning frameworks.
>
> Thank you for the insightful comment.   We want to clarify that we propose a parametric data augmentation, based on which we propose a general time series learning framework AutoTCL.  We verify the "generalization" with multiple experiments.
> 1.   We consider datasets from different domains, as recognized by the reviewer.
> 2.   We consider two different tasks, forecasting and classification. For forecasting, we consider both univariate and multivariate settings.  Results in Table 1,2,3,8,10,11,12 verify the advantage of AutoTCL over other contrastive learning baselines.
> 3.   We consider two state-of-the-art methods, CoST and TS2vec as backbones.  For each method, we use it in AutoTCL and compare it with basic augmentation, random augmentation, and adversarial augmentations. Results in 4,6,7, verify the advantage of parametric data augmentation.
>
> [1] Chen, Xinlei, and Kaiming He. "Exploring simple siamese representation learning." Proceedings of the IEEE/CVF conference on computer vision and pattern recognition. 2021.
> [2]Chen, Ting, et al. "Big self-supervised models are strong semi-supervised learners." Advances in neural information processing systems 33 (2020): 22243-22255.
>
> > W2. Discussion of Related Work: The authors have given sufficient discussion about augmentation techniques in the original data space. Recently, another branch of augmentations has stemmed from the feature space. I'd also suggested authors to discussion them for a more comprehensive review. [1,2,3]
>
>
> [1] Towards domain-agnostic contrastive learning
>
> [2] Metaug: Contrastive learning via meta feature augmentation
>
> [3] Hallucination Improves the Performance of Unsupervised Visual Representation Learning
>
>
> Thanks for your suggestion. We considered these papers and admitted their contribution to data augmentation. We added the discussion of these papers in the related work section.
>
>
> > Q1 The performance of Multivariate time series forecasting results seems to be sub-optimal compared with univariate time. Could the authors provide some insights?
>
> In a multivariate forecasting setting, we decrease the average MSE by 2.9% and the average MAE by 1.2% than CoST. In Table 2,10, our method gained comparable results and the most time of first/second place. Although the improvement is lower than that in the univariate setting, it is still significant. We think the potential reason is that feature-level correlation is crucial in multivariate forecasting, however, this is not the research focus of this paper.  We thank the reviewer for asking an interesting question.  We will explore this in future works, and study augmentations for multivariate settings, as the reviewer has suggested.
>
> Should you have any further questions or require additional information, please feel free to contact us.

---

> > ### Comment · Reviewer_NeuG · 2023-11-22
> >
> > I appreciate the authors' reply. I've also read other reviewers' comments. I'd raise the score from my side.

---

### Official Review · Reviewer_CECe · 2023-10-29

**Soundness:** 2 fair
**Presentation:** 2 fair
**Contribution:** 2 fair
**Rating:** 5
**Confidence:** 4

**Summary:**

The paper proposes a parametric augmentation method to automate the selection of augmentations in contrastive time-series representation learning. The method is evaluated on multiple datasets for the tasks of forecasting and classification.

**Strengths:**

The paper addresses an important topic in contrastive self-supervised learning: automating the selection of augmentations.

**Weaknesses:**

My main concern with the paper is that similar techniques were proposed earlier, such as Rommel et al., 2022. However, the paper does not provide any explanation beyond merely mentioning it in the related work section. It fails to clarify why the proposed method is different, necessary, or how it compares against Rommel et al., 2022. Additionally, there is no baseline for when augmentations are selected at random, for example, using RandAugment. Several other important prior works are missing as well, such as transformation prediction in HAR.

**Questions:**

Why is the evaluation for time-series classification limited to the UEA dataset? There are several other datasets available that cover modalities like EEG, ECG, and IMU. This choice is not very convincing.

Also, considering the success of masked autoencoders that do not require anchor/positive generation, what is the usefulness of such an approach? In most cases, the performance is close to that of InfoTS, and further hyperparameter tuning of InfoTS is likely to bridge the performance gap. I see very limited utility of the proposed method.

There is no silver bullet, so what are the limitations of the proposed method? The paper also mentions the incorporation of frequency domain augmentations as future work, but this has already been extensively explored in the literature.

---

> ### Author Response · Authors · 2023-11-21
> **Official response to Reviewer CECe(Part 1)**
>
> Dear reviewer CECe,
>
> thank you for taking the time to review our work and providing feedback. In the following, we aim to address your questions and concerns.
>
> > W1 My main concern with the paper is that similar techniques were proposed earlier, such as Rommel et al., 2022. However, the paper does not provide any explanation beyond merely mentioning it in the related work section. It fails to clarify why the proposed method is different, necessary, or how it compares against Rommel et al., 2022. Additionally, there is no baseline for when augmentations are selected at random, for example, using RandAugment. Several other important prior works are missing as well, such as transformation prediction in HAR.
>
> Thank you for your insightful feedback regarding the similarities between our work and Rommel et al., 2022. Our paper, while addressing some similar themes, fundamentally diverges from Rommel et al. in both approach and applicability. Our focus is on time series data augmentation from an information theory perspective, offering a broader scope than the EEG-specific context in Rommel et al.'s work. **We propose a novel factorization-based framework for guiding data augmentations in contrastive self-supervised learning, which is a significant departure from the extension of the AutoAugment framework employed by Rommel et al.** This framework, embodied in our contrastive learning framework with parametric augmentation (AutoTCL), is specifically designed to be more generalizable across different types of time series data.
>
> Moreover, our work is characterized by its empirical validation across a variety of time series forecasting contexts, demonstrating substantial improvements in performance metrics such as MSE, MAE, and classification accuracy. This level of empirical exploration and the performance improvements noted in our work are not paralleled in Rommel et al., which primarily focuses on EEG signal augmentation.
>
> The technical distinction, coupled with the broader applicability of our work beyond EEG data, underlines the unique contributions of our research to the field of contrastive learning with data augmentation for time series analysis. We believe these aspects collectively highlight the novelty and necessity of our approach in the evolving landscape of time series data analysis. We have added some discussions in the paper to make the difference more clear.
>
> We appreciate your suggestion to consider the methodologies employed in transformation prediction within Human Activity Recognition (HAR) as potentially relevant to our work. Indeed, the challenges faced in HAR, particularly in extracting and preserving meaningful features from complex time series data during transformations, resonate with the objectives of our research. We have included some related works from that topic.
>
> To address the reviewer's concern, we adopt a RandAugment, which is used in a famous contrastive learning framework SimCLR[1], and show the results in Tables 4, 6, 7.  We also show the summary results in the below. As the results suggested, our method outperforms SimCLR with random augmentations by large margins.
>
> | Dataset | Ours(MSE) |Ours(MAE)  | RandAugment(MSE) |RandAugment(MAE)  |
> | ------- | ----------| ----------|-------------|-------------|
> | ETTh1   | 0.076     |  0.207    | 0.112       | 0.254       |
> | ETTh2   | 0.158     |  0.299    | 0.168       | 0.321       |
> | ETTm1   | 0.046 	  |  0.154    | 0.065       | 0.187       |
> | Elec.   | 0.365     |  0.348    | 0.376       | 0.358       |
> | WTH     | 0.160     |  0.287    | 0.184       | 0.310       |
> | Lora    | 0.177     |  0.373    | 0.191       | 0.299       |
> | Avg.    | 0.157     |  0.258    | 0.176       | 0.286       |
>
>
>
> [1] Chen, Ting, et al. "A simple framework for contrastive learning of visual representations." International conference on machine learning. PMLR, 2020.

---

> ### Author Response · Authors · 2023-11-21
> **Official response to Reviewer CECe(Part 2)**
>
> > W2 Also, considering the success of masked autoencoders that do not require anchor/positive generation, what is the usefulness of such an approach? In most cases, the performance is close to that of InfoTS, and further hyperparameter tuning of InfoTS is likely to bridge the performance gap. I see very limited utility of the proposed method.
>
> Thank you for your valuable feedback. Next, we show the merits of AutoTCL, particularly in the context of the broader landscape of self-supervised learning techniques, including both contrastive learning and masked autoencoders.
>
> In this study, we intentionally focus on contrastive learning due to its distinct advantages and potential in handling complex time series data. While both contrastive learning and masked autoencoders are popular in self-supervised learning, we believe that the former offers unique opportunities, particularly in terms of creating robust embeddings and facilitating better understanding and interpretation of time series data. Our method, AutoTCL, leverages these strengths, providing a novel and effective way to apply contrastive learning to time series data.
>
> Besides, our proposed method introduces a novel factorization-based framework within the domain of contrastive learning. This approach, unlike traditional methods, does not rely on prefabricated knowledge, offering a more flexible and adaptable solution. The adaptability of AutoTCL to diverse time series instances are key differentiator in comparison to other self-supervised learning methods.
>
> Performance and Utility: AutoTCL shows promising results, comparable to existing methods like InfoTS, even without extensive hyperparameter optimization. In Table 9,10, we have done experiments on six different benchmark dataset and  the results shows that compared with InfoTS, our method reduced MSE by 16.5% , MAE by 5.8% in univariate setting, and MSE by 18.0% , MAE by 10.3% in the multivariate setting. This initial success underscores the potential of our approach. Beyond performance metrics, the adaptability and interpretability of AutoTCL make it a valuable tool for a range of practical applications.
>
>
> > There is no silver bullet, so what are the limitations of the proposed method? The paper also mentions the incorporation of frequency domain augmentations as future work, but this has already been extensively explored in the literature.
>
> Thanks for the reviewer's comment. A potential limitation of our method, AutoTCL, is that it does not explicitly consider the correlation between different features in time series data. This omission can be significant, especially in datasets where inter-feature relationships are crucial for understanding the underlying patterns and trends. Regarding the incorporation of frequency domain augmentations, we appreciate your pointing out the extensive exploration of this area in existing literature. Our mention of this as future work was intended to highlight our interest in integrating these well-established techniques with our novel AutoTCL framework. We believe that combining frequency domain augmentations with our factorization-based approach could offer new insights and potentially enhance the model's performance. We plan to conduct a thorough review of the existing literature in this area to inform our approach and ensure our contributions are novel and valuable.

---

> ### Comment · Reviewer_CECe · 2023-11-21
>
> Thank you for the clarification and detailed response.
>
> >regarding the similarities between our work and Rommel et al., 2022. Our paper, while addressing some similar themes, fundamentally diverges from Rommel et al. in both approach and applicability...
>
> But without comparison we will never know if the proposed method is better!
>
>  >our work is characterized by its empirical validation across a variety of time series forecasting contexts....
>
> The paper's focus is too heavy on forecasting dataset. The paper title should reflect this. I will update my score given the title clearly reflect what a reader should expect in the paper. Currently, the title is too generic!
>
> >The technical distinction, coupled with the broader applicability of our work beyond EEG data, underlines the unique contributions...
>
> Paper does not provide results on any EEG datasets, hard to say it goes beyond or not.
>
> >A potential limitation of our method, AutoTCL, is that it does not explicitly consider the correlation between different features in time series data...
>
> So this method may not work for multi-channel EEG or ECG signals without some modification and more suited for univariate signals. I think it is very important that title should be updated to show that this is about time series forecasting.

---

> > ### Author Response · Authors · 2023-11-21
> > **Further response to Reviewer CECe(Part 2/2)**
> >
> > > So this method may not work for multi-channel EEG or ECG signals without some modification and more suited for univariate signals. I think it is very important that title should be updated to show that this is about time series forecasting.
> >
> > Thanks for the interesting question. From our results in univariate and multivariate forecasting tasks, we observe that our method can improve by larger margins in the univariate setting. That's why we believe that considering the feature-level correlation in the parametric data augmentation is promising.  However, it doesn't mean that the proposed method may not work for multi-channel EEG or ECG signals. The reason is that our method is a general framework and can be combined with different backbones. For example, CoST, which is the default backbone, considers the feature level correlations.
> >
> > We thank the reviewer for the suggestions and recognize the author's concern about the adaptability of the proposed method.  In this study, we focus on general time series data and try to find a purely data-driven automatic data augmentation method for contrastive learning. On the one hand, this research problem is much more challenging than the setting in Rommel 22 paper, where they train the policy network with a supervised validation dataset and choose candidate data augmentations based on domain knowledge that is specific to EEG (Section 3). As a general framework,  in this paper, we don't include any classification label or prior knowledge. We believe that the research problem is significant and the proposed solution has more practical potential in applications where domain knowledge and labels are unavailable. On the other hand, we admit that it is also risky because time series data are very diverse. In some applications, for example, biosignals, domain knowledge is still necessary for model training and data augmentation. That is also a limitation of our work. We will include more discussions in the paper.
> >
> >
> >  Thanks again for your quick response.  We look forward to your further comments.

---

> ### Author Response · Authors · 2023-11-21
> **Further response to Reviewer CECe(Part 1/2)**
>
> Dear reviewer CECe,
>
> Thank you so much for your quick response. We are happy to provide more clarifications.
>
> > But without comparison we will never know if the proposed method is better!
>
> We got the reviewer's concern. However, it is unlikely to directly and fairly compare the proposed method with CADDA (the paper mentioned).  The reasons are:
>
> Based on our understanding, CADDA can only used in supervised time series classification, because it focuses on class-wise automatic data augmentation. The automatic augmentation is selected by a policy network, which is trained with a validation dataset.
>
> However, in our task, we focus on contrastive learning with automatic data augmentations.  In the literature, Contrastive Learning is usually considered a self-supervised learning method or an unsupervised learning method. (For further clarification, we have added "Self-supervised" to our title).
>
> We adopt the routinely adopted unsupervised settings to set up our experiments for a fair comparison [1][2][3][4]. The model doesn't have the class information and the validation datasets with labels. That is the reason why we acknowledge the importance and relevance of the mentioned paper but don't compare it with it.
>
> We thank the reviewer for the comment. We have modified the related work part and experimental setup part to explain according to our discussion.
>
>
> [1] Woo, Gerald, et al. "CoST: Contrastive Learning of Disentangled Seasonal-Trend Representations for Time Series Forecasting." International Conference on Learning Representations. 2022.
> [2] Yue, Zhihan, et al. "Ts2vec: Towards universal representation of time series." Proceedings of the AAAI Conference on Artificial Intelligence. Vol. 36. No. 8. 2022.
> [3] Luo, Dongsheng, et al. "Time series contrastive learning with information-aware augmentations." Proceedings of the AAAI Conference on Artificial Intelligence. Vol. 37. No. 4. 2023.
> [4] Xiang Zhang, Ziyuan Zhao, Theodoros Tsiligkaridis, and Marinka Zitnik. Self-supervised contrastive
> pre-training for time series via time-frequency consistency. Advances in Neural Information
> Processing Systems, 35:3988–4003, 2022.
>
> > The paper's focus is too heavy on forecasting dataset. The paper title should reflect this. I will update my score given the title clearly reflect what a reader should expect in the paper. Currently, the title is too generic!
>
> Thanks for the reviewers' comments. We agree that we focus on time series forecasting. The reason is that in this paper, we study the "parametric augmentation for time series contrastive learning". As we have discussed in the above comment.  Contrastive Learning is usually considered a self-supervised learning method or an unsupervised learning method. Generally, it learns an encoder to map the time series instance to a vector, as we introduced in  Section 3.1. Time series forecasting is the most popular application for time series representation. At the same time, researchers also observe that this framework can achieve promising results in classification datasets, but again, in the default setting, contrastive learning or self-supervised learning methods first learn representations in an unsupervised way. Then a supervised machine learning model like SVM or LR is used for classification.
>
> We thank the reviewer for the suggestion and have notified our paper's title by adding 'self-supervised'. That is "Parametric Augmentation for Self-Supervised Time Series Contrastive Learning" and add more clarification to distinguish our method from supervised ones.
>
> > Paper does not provide results on any EEG datasets, hard to say it goes beyond or not.
>
> We are sorry for this imprecise argument in our previous response.

---

> > ### Comment · Reviewer_CECe · 2023-11-21
> >
> > >We thank the reviewer for the suggestion and have notified our paper's title by adding 'self-supervised'. That is "Parametric Augmentation for Self-Supervised Time Series Contrastive Learning" and add more clarification to distinguish our method from supervised ones.
> >
> > It is even more generic in my opinion. There should be `forecasting` in the title. I leave the final decision to AC. I have raised my score based on discussion with the authors. Thank you for all the responses.

---

### Official Review · Reviewer_4aGN · 2023-10-30

**Soundness:** 3 good
**Presentation:** 3 good
**Contribution:** 2 fair
**Rating:** 6
**Confidence:** 4

**Summary:**

The study employs a parametric neural network to decompose time series into two components: the informative segment and the task-irrelevant segment. Through a data-driven approach, the method adaptively transforms input instances by generating masks to produce viable positive samples, ultimately enhancing the unsupervised performance in time series analysis.

**Strengths:**

1. The study is well-grounded in theory concerning its motivation and summarizes the theoretical conditions that the " GOOD VIEWS" of contrastive learning should meet.

2. The proposed method exhibits outstanding performance in experimental results.

**Weaknesses:**

The paper emphasizes the use of a parametric module to decompose time series into an informative part and a task-irrelevant part and perform parameter transformation on the informative part to obtain an enhanced view. However, it is not sufficiently clear.

1. What role does g play specifically? Especially in the analysis of the ablation experiments, I only observed differences in the results;

2. It's ambiguous why h is able to focus on the informative part of the sequence. The author should provide several case studies to prove that h can indeed play a role in separating the information part and noise part of the time series.

3. The two networks h and g use the same structure. How to ensure that they indeed perform the two different functions claimed by the author?

4. How is Δv set in the experiments?

5. The paper does not verify whether the positive view generated by the proposed method contains more information than the original sequence. Although the paper showcases the augmented instances through visualization in Figure 5, as the authors mentioned in the abstract, "it is impractical to visually inspect the temporal structures in time series." Providing explanations for the generated positive samples based on Property 3 or other relevant data would be beneficial.

6. From the visualization in Figure 5, it appears that the output mask of g is a binary mask, which does not match the description in the article. The article interprets g as a non-zero transformation mask. If the output of g is also a binary mask, does it mean that the optimal transformation method for sample enhancement that the model learns is simply to mask some observations of the original sequence?

**Questions:**

Please see the questions in the Weaknesses.

---

> ### Author Response · Authors · 2023-11-21
> **Official response to Reviewer 4aGN (Part 1)**
>
> Dear reviewer 4aGN,
>
> thank you for taking the time to review our work and providing feedback. In the following, we aim to address your questions and concerns.
>
> >W1 What role does g play specifically? Especially in the analysis of the ablation experiments, I only observed differences in the results;
>
> As we discussed in Section 3.2, $g$ is an invertible function to conduct data augmentation.  Intuitively, given an $x$, we use factorization function $h$ to detect the informative component, and then adopt $g$ to generate an augmented view for the following contrastive learning. We have added some discussion (Page 9) in the analysis of ablation experiments to address the reviewer's concern.
>
>
> >w2 It's ambiguous why h is able to focus on the informative part of the sequence. The author should provide several case studies to prove that h can indeed play a role in separating the information part and noise part of the time series.
>
> The function $h: \mathbb{R}^{T\times F} \rightarrow \\{0, 1\\}^{T\times 1}$ aims to detect the informative components of the input.  To encourage  $h$ to focus on the informative part of the sequence, we train the function $h$ with the principle of relevant information, which is shown in Eq. (6). $H(v^*)$ encourages reducing uncertainty and obtaining statistical regularity and the second part, $D(P_x||P_{v^*})$ is for preserving the descriptive power. After training, the generated $v^*$ will contain useful and compact information about the original sequence. Since $v^*$ is generated with an invertible function $g$ from $x^*$. That means the mapping between $x^*$ and $v^*$ is one by one, and $x^*$ also contains useful and compact information about the original sequence. In other words, $\mathbf{h}$ is able to focus on the informative part of the sequence. To empirically verify the claim, we conducted several case studies in Appendix D.3.5,  from the results, the informative parts detected by $h()$ appear to retain the prominent peaks and troughs of the original sequence, which are typically considered significant by domain experts. In time series analysis, such prominent features often correspond to critical events or changes in the underlying system's behavior. We have added some discussions to further clarify our observations (Page 22).
>
> > W3 The two networks $h$ and $g$ use the same structure. How to ensure that they indeed perform the two different functions claimed by the author?
>
> The factorization function $h$ aims to detect the informative component and $g$ aims to generate an augmented view. We believe that these two tasks are relevant and rely on the quality of hidden features. So we use a shared input layer and 1DCNN for both $h$ and $g$. However, $h$ and $g$ process the encoded features differently, through the factorization head and the transformation head, to produce the desired binary and non-zero outputs. factorization and transformation heads don't share parameters and they adopt different activation functions. Moreover, The network $h$ is trained with a size constraint loss function and the principle of relevant information, which encourages that its output is a compact binary vector. This binary vector represents a mask that identifies the informative components of the input. The size constraint is critical as it imposes sparsity on the output, guiding $h$ to select only the most salient features of the data. Conversely, $g$ is trained only with the principle of relevant information, generating a non-zero vector that enhances the original input for augmented views. Without the size constraint, $g$ is free to explore a wider space of augmentation possibilities.

---

> ### Author Response · Authors · 2023-11-21
> **Official response to Reviewer 4aGN (Part 2/2)**
>
> > W4 How is $\Delta v$ set in the experiments?
>
> As we introduced in section 3.3, Inspired by Dropout and TS2Vec, we implement the function $\eta(v^*,\Delta v)$  by randomly masking the hidden representation. Specifically, given a latent vector of a view $v$, after the first hidden layer, we randomly mask it along the time dimension with a binary vector. Each element is sampled independently from a Bernoulli distribution, Bern$(0.5)$.
>
> > W5. The paper does not verify whether the positive view generated by the proposed method contains more information than the original sequence. Although the paper showcases the augmented instances through visualization in Figure 5, as the authors mentioned in the abstract, "it is impractical to visually inspect the temporal structures in time series." Providing explanations for the generated positive samples based on Property 3 or other relevant data would be beneficial.
>
>
> The intuitive understanding of property 3 is that if the map between $x$ and $v$ is one-to-many, the generated view will contain more information compared to the raw input $x$. In other words, a good augmentation should preserve the underlying semantics that two different $x$ cannot map the same $v$.  To empirically show this property, we conducted case studies in Appendix D 3.2. We observe that AutoTCL can generate diverse views.  At the same time, they are closer to the original input instance compared to other input instances. We have added some discussion in the Appendix to make it clear.
>
> > W6 From the visualization in Figure 5, it appears that the output mask of g is a binary mask, which does not match the description in the article. The article interprets g as a non-zero transformation mask. If the output of g is also a binary mask, does it mean that the optimal transformation method for sample enhancement that the model learns is simply to mask some observations of the original sequence?
>
> We are sorry for this typo in the previous caption. It should be $h(x)$.  The output mask of $h(x)$ is a binary mask. we have fixed this typo and some imprecise description in the caption  (now, it is Table 9).
>
> Should you have any further questions or require additional information, please feel free to contact us.

---

> ### Author Response · Authors · 2023-11-21
> **A kind reminder**
>
> Dear reviewer 4aGN,
>
> we submitted the reply a few days ago. Now the deadline for public comment is approaching. We are keen to ensure that our revisions and responses align with your expectations and address all your concerns effectively. Please feel free to let us know if you have other questions.

---

### Official Review · Reviewer_3ku1 · 2023-10-31

**Soundness:** 3 good
**Presentation:** 3 good
**Contribution:** 3 good
**Rating:** 8
**Confidence:** 4

**Summary:**

In this paper, the authors introduced a comprehensive augmentation framework for contrastive self-supervised learning. The framework effectively tackles the challenge of time series data augmentation by leveraging principles from information theory. Additionally, the authors conducted experiments to validate the performance of the proposed framework.

**Strengths:**

- The proposed approach effectively deals with the data augmentation problem by unifying various methods into a comprehensive framework through the utilization of information theory.
- Not only has the effectiveness of the framework been theoretically proven from an information theory perspective, but it has also been extensively validated through empirical experiments.

**Weaknesses:**

- There might be some minor inconsistencies or gaps in the proof that require further attention to ensure its rigor. Such as, in the proof of Property 1. An invertible mapping is not necessarily a one-to-one mapping, which depending on the domain. Of course, this does not affect the subsequent proof.
- Some errors on formatting：page 6 “as random timestamp masking“, it seems unnecessary to bold it.
- Font size of some tables is too low.

**Questions:**

none

---

> ### Author Response · Authors · 2023-11-21
> **Official response to Reviewer 3ku1**
>
> Dear reviewer 3ku1,
>
> thank you for taking the time to review our work and providing feedback. In the following, we aim to address your questions and concerns.
>
> > w1. There might be some minor inconsistencies or gaps in the proof that require further attention to ensure its rigor. Such as, in the proof of Property 1. An invertible mapping is not necessarily a one-to-one mapping, which depending on the domain. Of course, this does not affect the subsequent proof.
>
> Dear reviewer 3ku1, thank you for recognizing our contributions.  However, we respectfully disagree with the comment.
> In fact, an invertible mapping, by definition, is necessarily a one-to-one (injective) mapping.
>
> To clarify:
>
> 1. **Invertible Mapping**: A function `f: A -> B` is invertible if there exists a function `g: B -> A` such that `g(f(a)) = a` for all `a` in `A` and `f(g(b)) = b` for all `b` in `B`. For a function to be invertible, it must be both injective (one-to-one) and surjective (onto).
>
> 2. **One-to-One (Injective) Mapping**: A function is one-to-one if it never maps distinct elements of its domain to the same element of its codomain. In other words, if `f(a) = f(b)` implies `a = b` for any `a, b` in the domain.
>
>
> 3. **Onto (Surjective) Mapping**: A function is onto if every element of the codomain is the image of at least one element of the domain.
>
> For a function to be invertible, it must be both one-to-one and onto. The one-to-one property ensures that the function doesn't map two different elements to the same value (necessary for the existence of a unique inverse mapping), and the onto property ensures that every element in the codomain has a pre-image in the domain (ensuring that the inverse mapping is well-defined for every element in the codomain).
>
> > w2. Some errors on formatting：page 6 “as random timestamp masking“, it seems unnecessary to bold it.
> > w3. Font size of some tables is too low.
>
> Thanks for the suggestions. We have revised our paper accordingly.
>
>
> Should you have any further questions or require additional information, please feel free to contact us.

---

### Official Review · Reviewer_3kKY · 2023-11-01

**Soundness:** 3 good
**Presentation:** 3 good
**Contribution:** 3 good
**Rating:** 8
**Confidence:** 4

**Summary:**

The paper starts by making the following observation: time series data is complex, high-dimensional, and harder to label than images or languages.
Deep learning requires lots of labeled data, but self-supervised learning offers a way to learn from unlabeled data. Applying general augmentation to diverse time series data is challenging; specific augmentations guided by domain knowledge are often needed.

Contributions:
- The paper introduces a factorization-based framework, AutoTCL, for adaptive data augmentation in time series contrastive learning. AutoTCL uses a neural network to factorize time series instances, preserving semantics, and optimizing against a contrastive loss.
- Empirical studies show the method outperforms benchmarks.

**Strengths:**

The paper has the following strenghts:
- It is clear, the writing is good.
- The empirical analysis seems sound. From table 4, there are benefits to the approach compared to relevant baselines and ablated versions of the model.
- The idea is proposed in a principled, justified way.

**Weaknesses:**

The paper's weaknesses are:
- Tables 6 and 7 along with the tables from the main paper seem to showcase marginal improvements over CoST, which was the architectural basis of their approach. In particular it is hard to determine if the difference is statistically significant.

- Most of the comparison in table 6/7 is less relevant than the ablation study. The reason is the following: the authors are comparing different architectures. Most of these perform less well than CoST, and they are building on top of CoST. Hence these results are quite expected and should in my opinion not be presented first.

- The authors propose a new augmentation scheme but focus only on CoST to showcase the performance of their approach. This means it is hard to determine whether their technique's benefits generalize to other settings.

- Many of the comparisons hinge on the average of performance on other datasets. However I have some concerns about Lora values being strongly different from other approaches (my 3rd question in the next section). This and the unweighted average of results could bias the conclusions somewhat.

**Questions:**

- Could the authors please provide experimental results on the other usual datasets, Illness / Exchange rate / Traffic?

- Could the authors please provide error bars/statistical tests for at least a subset of the experiments in tables 6/7? It is difficult to estimate whether the findings are statistically significant.

- In table 4, Lora is the only dataset for which CoST results from prior tables diverge strongly from w/o Aug. We would expect those two results to be quite close. Could the authors expand upon that point?

---

> ### Author Response · Authors · 2023-11-21
> **Official response to Reviewer 3kKY (Part 1)**
>
> Dear reviewer 3kKY,
>
> thank you for taking the time to review our work and providing feedback. In the following, we aim to address your questions and concerns.
>
> > W1 Tables 6 and 7 along with the tables from the main paper seem to showcase marginal improvements over CoST, which was the architectural basis of their approach. In particular it is hard to determine if the difference is statistically significant.
> > Q2. Could the authors please provide error bars/statistical tests for at least a subset of the experiments in tables 6/7? It is difficult to estimate whether the findings are statistically significant.
>
> Thanks for the suggestion.  We have conducted an extra experiment on the Electricity dataset.  As shown in the following table, the standard deviation for the univariate forecasting task in Electricity is smaller than 0.005, indicating that our method is robust to network initialization. We have similar observations in other datasets. Considering both univariate and multivariate settings, our method can outperform CoST in most cases. Thus, we believe that improving our method over CoST is significant.
>
> | $L_y$ | MSE                 | MAE                |
> | ----- | ----------------- | ----------------- |
> | 24    | 0.241 $\pm$ 0.001	  | 0.264 $\pm$ 0.002  |
> | 48    |  0.286 $\pm$  0.002 | 0.293  $\pm$  0.002|
> | 168   | 0.394 $\pm$ 0.002	  | 0.365 $\pm$  0.003 |
> | 336   | 0.546 $\pm$  0.004  | 0.462 $\pm$  0.004 |
>
>
> > W2. Most of the comparison in table 6/7 is less relevant than the ablation study. The reason is the following: the authors are comparing different architectures. Most of these perform less well than CoST, and they are building on top of CoST. Hence these results are quite expected and should in my opinion not be presented first.
>
>
> Thanks for the suggestion.  We revised our paper accordingly to move tables `Ablation studies using CoST backbone` and `Ablation studies using TS2Vec backbone` in front of comparison to other baselines in the Appendix.
>
> > W3. The authors propose a new augmentation scheme but focus only on CoST to showcase the performance of their approach. This means it is hard to determine whether their technique's benefits generalize to other setting.
>
> Thanks the reviewer for the suggestion. Note that, we do consider multiple settings to comprehensively evaluate the proposed techniques.
> - We use datasets from different domains.
> - We consider two different tasks, forecasting and classification. For forecasting, we consider both univariate and multivariate settings.
> - We consider two state-of-the-art methods, CoST and TS2vec as backbones. For each method, we use it in AutoTCL and compare it with basic augmentation, random augmentation, and adversarial augmentations.
>
> We agree that a powerful backbone is necessary to achieve competitive performance. Thus, by default, we choose CoST as the backbone for the forecasting task and  TS2Vec for the classification task.
>
> > W4. Many of the comparisons hinge on the average of performance on other datasets. However I have some concerns about Lora values being strongly different from other approaches (my 3rd question in the next section). This and the unweighted average of results could bias the conclusions somewhat.
>
> Thanks for the suggestion. We have added more analysis in Section 4.1 besides the average results. Compared to CoST,  AutoTCL decreases both MAE and MSE in all datasets in the univariate setting. In the multivariate setting, AutoTCL outperforms CoST in 7 cases.

---

> > ### Comment · Reviewer_3kKY · 2023-11-22
> > **Response to author's rebuttal**
> >
> > I thank the authors for responding to my review. I have read their response in detail, and feel that they address the main concerns I had, in particular by adding more empirical results strenghtening their initial conclusions.
> >
> > I believe the paper should be accepted in this state, and am raising my score to an 8.

---

> ### Author Response · Authors · 2023-11-21
> **Official response to Reviewer 3kKY (Part 2)**
>
> > Q1. Could the authors please provide experimental results on the other usual datasets, Illness / Exchange rate / Traffic?
>
> We have done the experiments on the Traffic dataset. The following are the results of univariant setting based on CoST backbone. We observe consistent improvement of our methods over two leading baselines. We will include that in the updated version.
>
> | $L_y$ | Ours(MSE)     | CoST(MSE)    | TS2Vec(MSE)         |
> | ----- | -----------------  | ---------------|--------------|
> | 96    | 0.253     	     | 0.284          | 0.431 |
> | 192   | 0.271              | 0.302          | 0.437 |
> | 336   | 0.312 	         | 0.340          | 0.453 |
> | 720   | 0.357              | 0.390          | 0.464 |
>
>
> | $L_y$ | Ours(MAE)     | CoST(MAE)    | TS2Vec(MAE)         |
> | ----- | -----------------  | ---------------|--------------|
> | 96    | 0.353     	     | 0.379          | 0.484 |
> | 192   | 0.373              | 0.398          | 0.489 |
> | 336   | 0.414 	         | 0.435          | 0.500 |
> | 720   | 0.447              | 0.474          | 0.508 |
>
> The following are the results of multivariant setting.
>
> | $L_y$ | Ours(MSE)     | CoST(MSE)    | TS2Vec(MSE)         |
> | ----- | -----------------  | ---------------|--------------|
> | 96    | 0.715     	     | 0.759          | 1.038 |
> | 192   | 0.722              | 0.757          | 1.042 |
> | 336   | 0.730 	         | 0.765          | 1.064 |
> | 720   | 0.746              | 0.784          | 1.085 |
>
>
> | $L_y$ | Ours(MAE)     | CoST(MAE)    | TS2Vec(MAE)         |
> | ----- | -----------------  | ---------------|--------------|
> | 96    | 0.396     	     | 0.442          | 0.574 |
> | 192   | 0.396              | 0.434          | 0.588 |
> | 336   | 0.396 	         | 0.435          | 0.594 |
> | 720   | 0.403              | 0.444          | 0.604 |
>
>
> > Q3. In Table 4, Lora is the only dataset for which CoST results from prior tables diverge strongly from w/o Aug. We would expect those two results to be quite close. Could the authors expand upon that point?
>
> The Lora dataset is a newly introduced real-world dataset that captures the wireless signal data using the LoRa devices. It contains 74 days of data with timestamps. The feature contains the signal-to-noise ratio(SNR), Received Signal Strength Indicator(RSSI), Bit Error Rate(BER), and Packet Reception Rate(PRR).   A main problem for forecasting is the huge noise due to environmental changes, making it necessary to conduct data augmentation to make the model more robust. That is the reason for the big gap between w/ and w/o Aug.

---

### Meta-Review · Area_Chair_8xJQ · 2023-12-02

**Metareview:**

**Summary:** The paper highlights the need for labeled data in deep learning and the potential of self-supervised learning from unlabeled data.  The authors introduce AutoTCL, a factorization-based framework that applies adaptive data augmentation in time series contrastive learning.  Experimental studies demonstrate that AutoTCL outperforms benchmark methods.  The proposed framework leverages principles from information theory and automates the selection of augmentations to enhance unsupervised performance in time series analysis. AutoTCL is encoder-independent and compatible with various backbone encoders.  There is an agreement that the paper makes contributions to the field and that should be accepted.

**Strengths:** The paper provides sound empirical experiments and analysis, along with theoretical results, supporting its claims. The justifications for the proposed framework are well-established and logical. The study adopts a unique approach by incorporating data augmentation from the perspective of information theory.

**Weaknesses:** The reviewers' concerns are about the similarity of the proposed techniques to earlier approaches and the absence of baseline comparisons. The limited experiments and discussion with related work were also seen as areas for improvement. However, the authors have addressed these concerns and received positive feedback from the reviewers.  Moreover, Reviewer CECe highlights the need to adjust the title to be more in line with the work presented in the paper.

**Justification For Why Not Higher Score:**

The paper received a poster score due to issues regarding the lack of clear justifications for the design decisions of the method. Additionally, there is a need for more extensive results to compare the proposed method with other contrastive learning frameworks. The presentation of the paper primarily focuses on forecasting, but it is presented as a general method, so aligning the scope and presentation is necessary.

**Justification For Why Not Lower Score:**

The reviewers unanimously agree that the paper makes substantial contributions to the field despite the identified shortcomings. Therefore, it is not a paper that should be rejected.

---

### Decision · Program_Chairs · 2024-01-16

Accept (poster)